# Quantum Black Holes in Conformal Dilaton–Higgs Gravity on Warped Spacetimes

Reinoud Jan Slagter 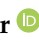

Asfyon, Astronomisch Fysisch Onderzoek Nederland, Institute for Theoretical Physics, University of Amsterdam, 1405EP Bussum, The Netherlands; info@asfyon.com; Tel.: +31-643900550

**Abstract:** A promising method for understanding the geometric properties of a spacetime in the vicinity of the horizon of a Kerr-like black hole can be developed by applying the antipodal boundary condition on the two opposite regions in the extended Penrose diagram. By considering a conformally invariant Lagrangian on a Randall–Sundrum warped five-dimensional spacetime, an exact vacuum solution is found, which can be interpreted as an instanton solution on the Riemannian counterpart spacetime, $\mathbb{R}^2_+ \times \mathbb{R}^1 \times S^1$, where $\mathbb{R}^2_+$ is conformally flat. The antipodal identification, which comes with a CPT inversion, is par excellence, suitable when quantum mechanical effects, such as the evaporation of a black hole by Hawking radiation, are studied. Moreover, the black hole paradoxes could be solved. By applying the non-orientable Klein surface, embedded in $\mathbb{R}^4$, there is no need for instantaneous transport of information. Further, the gravitons become "hard" in the bulk, which means that the gravitational backreaction on the brane can be treated without the need for a firewall. By splitting the metric in a product $\omega^2 \tilde{g}_{\mu\nu}$, where $\omega$ represents a dilaton field and $\tilde{g}_{\mu\nu}$ the conformally flat "un-physical" spacetime, one can better construct an effective Lagrangian in a quantum mechanical setting when one approaches the small-scale area. When a scalar field is included in the Lagrangian, a numerical solution is presented, where the interaction between $\omega$ and $\Phi$ is manifest. An estimate of the extra dimension could be obtained by measuring the elapsed traversal time of the Hawking particles on the Klein surface in the extra dimension. Close to the Planck scale, both $\omega$ and $\Phi$ can be treated as ordinary quantum fields. From the dilaton field equation, we obtain a mass term for the potential term in the Lagrangian, dependent on the size of the extra dimension.

**Keywords:** conformal invariance; dilaton field; black hole paradoxes; brane-world models; antipodal mapping; Klein surface; quintic polynomial; Hawking radiation

**PACS:** 04.20.−q; 04.50.−h; 04.62.+v; 02.40.−k; 04.70.Dy; 02.40.Dr; 03.65.Ud; 03.65.Ta; 03.65.Vf; 02.30.Jr

## 1. Introduction

One of the most profound unsolved problems in theoretical physics is the discrepancy between quantum mechanics (QM) and general relativity theory (GRT). One would like to obtain a consistent theory of quantum gravity, a major goal of theoretical physics (the "holy grail"), which would reconcile QM and Einstein's GRT. Over the last few decades, several authors have proposed different approaches in order to tackle this epic problem. The main question now is how to embark on this immense task. In general, there are two possible routes. One can modify general relativity theory and consider quantized field theory as the principal theory. An alternative would be to modify quantum theory and retain general relativity theory as the principal model. Before arriving at a decision, one might wonder where experimental tests and potential unification could take place. This is likely to occur at a physical location, where the energy density is huge. The physics in the vicinity of the horizon of a black hole fulfills this criterion. Gravitational radiation and Hawking radiation produced at this location could harbor valuable information within themselves.

Pioneering work on this problem was delivered by Hawking [1], who investigated particle creation near the horizon of a black hole. He found that black holes radiate as black bodies with a thermal spectrum. So, the black hole would be in a mixed state. However, this seems to violate a unitary evolution, because all information connected with the infalling particle ("in-state") appears to have been lost after the evaporation of the black hole. In quantum field theory (QFT) one usually deals with pure quantum states that evolve unitarily.

A possible solution was addressed by Almheiri et al. [2] to solve the conflict between complementarity and firewalls. The infalling observer would burn up at the horizon's firewall. However, this viewpoint conflicts with GRT. A related problem is how to interpret the induced regions in the Penrose diagram of the maximally extended Schwarzschild black hole solution in Kruskal–Szekeres coordinates. Maldacena et al. [3] proposed the "ER = EPR" hypothesis. The two exterior regions should be connected by an Einstein–Rosen bridge (or "wormhole"), and the connectivity in this spacetime is equivalent to quantum entanglement. However, one ignores the gravitational interaction between the two exterior regions I and II of the Penrose diagram. Another interesting method for solving the information paradox for certain regions is the so-called "island recipe" [4–6]. The method consists of a semi-classical framework of gravity while maintaining unitary black hole evaporation. In fact, the method solves the missing element in Hawking's original calculation. However, boundary unitarity is not sufficient, even in the restricted context of AdS/CFT. One proves that Hawking radiation continues to be in a thermal state (up to greybody factors) that is purified by interior modes, even at later times in the evaporation process. By using an entangled wedge construction, one shows that at later times, the interior degrees of freedom are not microscopically independent of the early Hawking radiation. This is essentially a notion of the complementarity. One identifies parts of the interior that are encoded in the Hawking radiation. In this method, to calculate the von Neumann entropy, one uses the formation of saddle points, determined by the "island" prescription, i.e., an arbitrary set of intervals on a Cauchy surface. The most striking feature is that the entanglement effect of the island can be accounted for in terms of its mirror image in the outgoing radiation.

Finally, we mention that some physicists have suggested that region II could represent another universe [7]. Even more exotic solutions have been proposed [8]. In this manuscript, we will focus on a totally different approach for the interpretation of the geometry near black holes, that is, the antipodal boundary condition, introduced in 1957 by Schrödinger (the so-called "elliptic interpretation" of the spacetime [9]). 't Hooft recently improved the method to solve several black hole paradoxes [10]. In brief, antipodal points on the horizon are identified. The spacetime inside the horizon is removed through the identification of the antipodes.

The approach we will address in this manuscript relies on the antipodicity of a conformally invariant warped 5D spacetime, following the approach of Randall–Sundrum (RS) [11,12]. These warped brane-world models provide a simplification of the full-string model. Only gravity can propagate into the bulk, while the other fields reside on the brane. A possible effective theory can then be constructed without a UV cutoff because the fundamental scale can be much smaller than the effective Planck scale. This model can also be applied to a Friedmann–Lemaître–Robertson–Walker (FLRW) spacetime [13]. Further, we will replace the antipodal boundary condition, which could be visualized by a non-orientable Möbius surface in $\mathbb{R}^3$, with a non-orientable Klein surface, which can be embedded in $\mathbb{R}^4$ [14,15]. This is mandatory because we work in a 5D spacetime. Many approximation procedures can be applied near the horizon to describe the gravitational interaction, i.e., the emission of gravitational waves, together with Hawking radiation. The calculations of the quasi-normal modes were conventionally performed on a fixed background, which is not adequate for the black hole evaporation process. This is a time-dependent process and one would like to obtain a complete scattering description for both the external and local observers, preferably by using dynamically differential equations. The method of 't Hooft bypasses the troublesome Planck area by using the Shapiro

time-delay effect [16,17]. The high-energy effects are replaced with high wave numbers of the spherical harmonics. In this manuscript, we will use the conformally invariant Lagrangian. The complementarity issue can then be reformulated by introducing a dilaton $\omega$, by $g_{\mu\nu} = \omega^{4/(d-2)}\tilde{g}_{\mu\nu}$ in d-dimensions [18]. The conformal invariance will spontaneously be broken by a mass term in the Lagrangian. The dilaton determines the scale. Different observers will then experience different scales. For very small scales, close to the Planck scale, it could be treated as a quantum field, similar to a scalar field. One observer sees Hawking radiation as matter, while the other one sees it as part of their vacuum. The approach in this manuscript is not the ultimate description of the behavior of a quantum black hole. The main reason for the antipodal approach is the remarkably consistent extension to the warped 5D spacetime by using the non-orientable Klein surface instead of the non-orientable Möbius surface. The topology of spacetime again becomes regular and has some advantages, as we shall see. There are, of course, other approaches. In a recent work by Penington [4], the "entanglement wedge reconstruction" of highly mixed states is applied. He argues that before the evaporation begins, a single, state-independent interior reconstruction exists for any code space of microstates with entropy strictly less than the Bekenstein–Hawking entropy. The firewall paradox could also be solved using this approach.

This manuscript contains the following sections. In Section 2, we formulate the model under consideration. In Section 3, we add a scalar field to the model, and in Section 4, we apply the model to the antipodal boundary condition.

## 2. The Model

It is conjectured that a conformally invariant theory of gravity promises interesting results when applied to the quantum-gravity area, particularly the conformal dilaton gravity (CDG) model [18–21]. We shall see that it is a route to constructing a topologically regular theory of gravity. Examples of topological regular solutions in field theory are solitons and instantons. We will encounter both solutions in due course. The singularities that arise in these solutions can be handled through suitable coordinate transformations. The topology dictates these mappings. The antipodal mapping meets the criteria, such as smoothness. Further, regularity also means the stability of the solution in the applied topology. This is rather difficult in more general settings. In Euclidean spaces, this is no problem [22,23]. Another topological feature is the contractibility of closed surfaces in a manifold. The Klein bottle, for example, must be embedded in $(4 + 1)$-dimensional spacetime to avoid self-intersection (Section 5). A theory is called conformally invariant at the classical level if its action is invariant under the conformal group of translations, dilatations, Lorentz transformations (LT), and special conformal transformations [23]. This is a local symmetry if the metric is dynamical, as will be considered here[1]. Let us consider the conformal invariant Lagrangian

$$S = \int d^d x \sqrt{-\tilde{g}} \left[ -\frac{1}{2}\xi(\Phi\Phi^* + \omega^2)\tilde{R} - \frac{1}{2}\tilde{g}^{\mu\nu}\left(\mathcal{D}_\mu\Phi(\mathcal{D}_\nu\Phi)^* + \partial_\mu\omega\partial_\nu\omega\right) \right.$$
$$\left. -\frac{1}{4}F_{\alpha\beta}F^{\alpha\beta} - V(\Phi,\omega) - \Lambda\kappa^{\frac{4}{d-2}}\xi^{\frac{d}{d-2}}\omega^{\frac{2d}{d-2}} \right], \tag{1}$$

which is invariant under

$$\tilde{g}_{\mu\nu} \to \Omega^{\frac{4}{d-2}}\tilde{g}_{\mu\nu}, \quad \omega \to \Omega^{-\frac{d-2}{2}}\omega, \quad \Phi \to \Omega^{-\frac{d-2}{2}}\Phi, \tag{2}$$

for a suitable choice of $V$ (for example, the simplest one, $V = 0$). Further, $\xi = (d-2)/4(d-1)$. In the vacuum case (Section 2.1), the original Einstein–Hilbert action is replaced with

$$S = \int d^4 x \frac{\sqrt{-\tilde{g}}}{2\kappa^2}\left(-\frac{1}{12}\omega^2\tilde{R} - \frac{1}{2}\tilde{g}^{\mu\nu}\partial_\mu\omega\partial_\nu\omega - \frac{1}{36}\Lambda\omega^4\right), \tag{3}$$

with $\kappa = 8\pi G_N$. Further, we will rescale $\omega^2 \to -6\frac{\omega^2}{\kappa^2}$ to store $\kappa$ in the last term in Equation (3). One preserves the unitarity and positivity properties when including a scalar field in the Lagrangian. The functional integral over $\omega$ now has to go over a contour parallel to the imaginary axis, a well-known feature of canonical quantum gravity, especially when investigating the Euclidean counterpart model using the Wick rotation (see Section 2.4). Note that we get back Newton's constant when conformal invariance is broken by the incorporation of matter fields, such as $\sim \sqrt{-g}m^2\Phi^2$ [18]. In fact, all physical constants eventually emerge as dimensionless combinations of Newton's constant. The energy-momentum tensor consists of the contribution from $\omega$, and in the non-vacuum case, the matter fields. In the resulting equations, we have the covariant derivative $\tilde{\nabla}$ with respect to the "unphysical" $\tilde{g}_{\mu\nu}$, which is defined in the CDG model initiated by 't Hooft [18], by writing

$$g_{\mu\nu} = \omega^{4/(d-2)}\tilde{g}_{\mu\nu}. \tag{4}$$

The gauge-covariant derivative is $\mathcal{D}_\mu\Phi = \tilde{\nabla}_\mu\Phi + ieA_\mu\Phi$ and $F_{\mu\nu}$ is the abelian field strength. A mass term in the expression of the potential will break the tracelessness of the energy-momentum tensor and thus the conformal invariance. This spontaneous breaking can be compared with the famous Brout–Englert–Higgs (BEH) mechanism. On small scales, the dilaton approaches zero, and no singularity occurs in this limit. We parameterize the scalar field and gauge field as

$$A_\mu = \left[0, 0, 0, \frac{1}{e}(P(t,r) - n)\right], \quad \Phi = \eta X(t,r)e^{in\varphi}, \tag{5}$$

where $n$ is the winding number, $\eta$ is the vacuum expectation value, and $e$ is the electric charge. The metric $\tilde{g}_{\mu\nu}$ is actually a meta-tensor. All the scale dependencies are contained in the dilaton and will be handled on equal footing with the scalar field and can be extended to scales close to the Planck scale. The reason for the use of a scalar-gauge field $(\Phi, A_\mu)$ in the action finds its origin in superconductivity, where the famous Meissner effect can be explained by the symmetry breaking below a critical temperature. In gravity theory, a comparable symmetry is desirable. It is commonly believed that the universe underwent a violent phase transition during an early epoch. A natural step is then to incorporate a matter term in $\Phi$ and $A_\mu$ on the right-hand side of the Einstein equations to describe the symmetry breaking in a common symmetry group [24]. This procedure was, after all, also successful for the electro-weak unification $(SU(2) \times U(1))$ and the unification with chromodynamics $(SU(3))$.

We consider here the Kerr-like spacetime on a warped 5D spacetime with $\mathbb{Z}_2$-symmetry [25,26]

$$ds^2 = \omega(t,r,y)^{4/3}y_0\left[-N(t,r)^2dt^2 + \frac{1}{N(t,r)^2}dr^2 + dz^2 + r^2(d\varphi + N^\varphi(t,r)dt)^2 + dy_5^2\right], \tag{6}$$

where $y_5$ is the extra dimension (not to be confused with the Cartesian $y^2$). Here, $\omega$ is called a "warp factor" in the formulation of RS 5D warped spacetime with one large extra dimension and negative bulk tension. It turns out that one can write $\omega(t,r,y_5) = \omega_1(t,r)\omega_2(y_5)$, with $\omega_2(y_5) = y_0 = $ constant (the length scale of the extra dimension). In an FLRW metric, one then recovers the well-known RS warp factor for $\omega_2(y_5)$ [13]. All the fields of the standard model reside on the brane, while gravity can also propagate in the extra dimension. The model possesses $\mathbb{Z}_2$-symmetry. One can formulate the model covariantly, which means that one should solve the 5D field equations simultaneously with the effective 4D field equations [27,28]. We do not consider matter fields in the bulk here.

The Einstein field equations become [25,26]

$$^{(5)}G_{\mu\nu} = -\Lambda_5\,^{(5)}g_{\mu\nu}, \tag{7}$$

$$^{(4)}G_{\mu\nu} = -\Lambda_{eff}\,^{(4)}g_{\mu\nu} + \kappa_4^2\,^{(4)}T_{\mu\nu} + \kappa_5^4\mathcal{S}_{\mu\nu} - \mathcal{E}_{\mu\nu}, \tag{8}$$

where we have written

$$^{(5)}g_{\mu\nu} = {}^{(4)}g_{\mu\nu} + n_\mu n_\nu, \tag{9}$$

with $n^\mu = [0,0,0,0,\frac{1}{\sqrt{y_0}}]$ being the unit normal to the brane. Here, $^{(4)}T_{\mu\nu}$ is the energy-momentum tensor on the brane, which also contains the contribution from the dilaton

$$^{(4)}T_{\mu\nu} = T^{(\omega)}_{\mu\nu} + T^{(scalar)}_{\mu\nu} \tag{10}$$

and $\mathcal{S}_{\mu\nu}$ is the quadratic contribution of the energy-momentum tensor $^{(4)}T_{\mu\nu}$ arising from the extrinsic curvature terms in the projected Einstein tensor. Further,

$$\mathcal{E}_{\mu\nu} = {}^{(5)}C^\alpha_{\beta\rho\sigma} n_\alpha n^{\rho\,(4)} g^\beta_\mu {}^{(4)}g^\sigma_\nu, \tag{11}$$

represents the projection of the bulk Weyl tensor orthogonal to $n^\mu$. It contains the information of the gravitation field in the bulk [25]. In the dynamical situation, in general, one needs to specify the initial data on the brane. Further, we have the conservation equation $\nabla^\mu \mathcal{E}_{\mu\nu} = \kappa^4_5 \mathcal{S}_{\mu\nu}$. It expresses the exchange of so-called Kaluza–Klein (KK) modes, i.e., the effective 4D modes of the 5D graviton [28].

Now, we again replace

$$^{(4)}\tilde{g}_{\mu\nu} \to \bar{\omega}^{2\,(4)}\tilde{g}_{\mu\nu}. \tag{12}$$

A variation of the action Equation (1) for a general $V^3$ with respect to $\tilde{g}_{\mu\nu}$, $\Phi$, and $\omega$ yields ($d = 4, 5$)

$$\xi \bar{\omega} \tilde{R} - \tilde{g}^{\mu\nu} \tilde{\nabla}_\mu \tilde{\nabla}_\nu \bar{\omega} - \frac{\partial V}{\partial \bar{\omega}} = 0, \tag{13}$$

$$\xi \Phi \tilde{R} - \tilde{g}^{\mu\nu} \mathcal{D}_\mu \mathcal{D}_\nu \Phi - \frac{\partial V}{\partial \Phi^*} = 0, \tag{14}$$

and

$$(\bar{\omega}^2 + \Phi\Phi^*)\tilde{G}_{\mu\nu} = T^{(\Phi)}_{\mu\nu} + T^{(\omega)}_{\mu\nu} - \tilde{g}_{\mu\nu} V - (\bar{\omega}^2 + \Phi\Phi^*)\mathcal{E}_{\mu\nu}\delta^4_d, \tag{15}$$

with

$$T^{(\omega)}_{\mu\nu} = \tilde{\nabla}_\mu \tilde{\nabla}_\nu \bar{\omega}^2 - \tilde{g}_{\mu\nu} \tilde{\nabla}^2 \bar{\omega}^2 + \frac{1}{\xi}\left(\frac{1}{2}\tilde{g}_{\alpha\beta}\tilde{g}_{\mu\nu} - \tilde{g}_{\mu\alpha}\tilde{g}_{\nu\beta}\right)\partial^\alpha \bar{\omega}\partial^\beta \bar{\omega}, \tag{16}$$

and

$$T^{(\Phi)}_{\mu\nu} = \mathcal{D}_\mu \mathcal{D}_\nu \Phi\Phi^* - \tilde{g}_{\mu\nu} \mathcal{D}^\alpha \mathcal{D}_\alpha \Phi\Phi^* + \frac{1}{\xi}\left(\frac{1}{2}\tilde{g}_{\alpha\beta}\tilde{g}_{\mu\nu} - \tilde{g}_{\mu\alpha}\tilde{g}_{\nu\beta}\right)\mathcal{D}^\alpha \Phi \mathcal{D}^\beta \Phi^*. \tag{17}$$

For $n = 4$, in Equation (15), there is the contribution from the projected Weyl tensor $\mathcal{E}_{\mu\nu}$.

In the vacuum case, the dilaton equation (13) delivers [25] no new information, as expected. After all, it is part of the gravitational sector. When the scalar field is incorporated, there will be an interaction between $\omega$ and $\Phi$ and some constraint on $V$. One observes that $\Phi$ and $\omega$ can be treated on equal footing.

### 2.1. The Vacuum Case

In a former study, in the case without a scalar field, we found an exact solution of the vacuum equations [25]. By isolating the PDEs for $\omega$, $N$, and $N^\varphi$ one can simultaneously solve the 5D Einstein equations, together with the effective 4D Einstein equations. By eliminating the equations for $N^\varphi$ from the 5D Einstein equations, it turns out that the equation for $\omega$ becomes

$$\ddot{\omega} = \frac{\dot{\omega}\dot{N}}{N} + \frac{1}{3}\frac{\dot{\omega}^2}{\omega} - \frac{N^2}{16}\sqrt[3]{18}\Lambda\kappa^{4/3}\omega^{7/3}y_0 - N^4\left(\frac{\omega' N'}{N} + \frac{4}{3}\frac{\omega'^2}{\omega} + \frac{1}{2}\frac{\omega'}{r}\right). \tag{18}$$

This equation, together with the equation for $\ddot{N}$ (see Equation (22)), must be solved numerically, which we will carry out in Section 3 when a scalar field is incorporated. There is, however, still an equation left

$$\bar{\omega}'' = -\frac{2n}{n-2}\frac{\Lambda y_0 \kappa^{\frac{4}{(n-2)}}\xi^{\frac{n-2}{4(n-1)}}\bar{\omega}^{\frac{n+2}{n-2}}}{N^2} - \frac{\bar{\omega}'N'}{N} - \frac{\bar{\omega}'}{2r} + \frac{4}{n-2}\frac{\dot{\omega}^2}{\bar{\omega}N^4} - \frac{\dot{\omega}\dot{N}}{N^5}. \tag{19}$$

When we eliminate $\dot{N}$ from the $\omega$ equation with the help of this equation, we surprisingly obtain,

$$\ddot{\omega} = -N^4\omega'' + \frac{5}{3\omega}\left(N^4\omega'^2 + \dot{\omega}^2\right). \tag{20}$$

This also happens for the effective 4D equations (8), which contain the extra term $\mathcal{E}_{\mu\nu}$, which is remarkable. The only explanation will be that this solution represents an instanton (see Section 2.4) in the warped spacetime. In this case, one can write the equations for $\omega, \bar{\omega}$, and $N$ for $n = 4, 5$

$$\ddot{\omega} = -N^4\omega'' + \frac{n}{\omega(n-2)}\left(N^4\omega'^2 + \dot{\omega}^2\right), \tag{21}$$

$$\ddot{N} = \frac{3\dot{N}^2}{N} - N^4\left(N'' + \frac{3N'}{r} + \frac{N'^2}{N}\right)$$
$$-\frac{n-1}{(n-3)\omega}\left[N^5\left(\omega'' + \frac{\omega'}{r} + \frac{n}{2-n}\frac{\omega'^2}{\omega}\right) + N^4\omega'N' + \dot{\omega}\dot{N}\right]. \tag{22}$$

Note that we are dealing with two different dilaton fields, $\omega$ and $\bar{\omega}$. For $n = 5$, we obtain the solution for $\omega$, and for $n = 4$, the solution for $\bar{\omega}$. This is due to the fact that we have two different local observers: one in the bulk and one in the brane. They use different notions of scale. The equation for $N^\varphi$ decouples. One obtains the solution

$$\omega = \left(\frac{a_1}{(r+a_2)t + a_3 r + a_2 a_3}\right)^{\frac{3}{2}}, \quad \bar{\omega} = \left(\frac{a_1}{(r+a_2)t + a_3 r + a_2 a_3}\right)$$
$$N^2 \equiv \frac{N_1(r)}{N_2(t)} = \frac{1}{5r^2}\frac{10a_2^3 r^2 + 20a_2^2 r^3 + 15a_2 r^4 + 4r^5 + C_1}{C_2(a_3+t)^4 + C_3}, \tag{23}$$

where $a_i$ and $C_i$ are some constants[4].

The result is that both partial differential equations are of the elliptic type. If we do not eliminate $\dot{N}$, the differential equations are only solvable numerically. The solutions for the two dilaton fields, $\omega$ and $\bar{\omega}$, differ only by the different exponents $\frac{3}{2}$ and 1, respectively. The solution for the metric component is the same (apart from the constants). Further, for the effective 4D solution, it turns out that $\frac{dN}{dt} = N^2 \mathcal{J}(t,r)$. So, for the location of the singularities, $\frac{dN}{dt} = 0$ holds. Further, $\frac{d^2\omega}{dtdr} \to \infty$ for $r = -a_2, t = -a_3$ and the zeros of the nominator and denominator of $N^2$. The solution for the angular momentum component is

$$N^\varphi = F_n(t) + \int \frac{1}{r^3\bar{\omega}^{\frac{n-1}{n-3}}}dr. \tag{24}$$

The Ricci scalar for $\tilde{g}_{\mu\nu}$ is given by

$$\tilde{R} = \frac{12}{N^2}\left[\dot{\bar{\omega}}^2 - N^4\bar{\omega}'^2\right], \tag{25}$$

which is consistent with the null condition for the two-dimensional $(t, r)$ line element, when $\tilde{R} = 0$. One can easily check that the trace of the Einstein equations is zero. Note that $N^2$ can be written as

$$N^2 = \frac{4\int r(r+a_2)^3 dr}{r^2[C_2(a_3+t)^4 + C_3]}. \tag{26}$$

The non-local conservation equations become [27]

$$\bar{\nabla}^{\mu} \mathcal{E}_{\mu\nu} = \bar{\nabla}^{\mu} \left[ \frac{1}{\bar{\omega}^2} \left( -\Lambda \kappa^{2\,(4)} \bar{g}_{\mu\nu} \bar{\omega}^4 + {}^{(4)}T_{\mu\nu}{}^{(\bar{\omega})} \right) \right],$$ (27)

So, the divergence of $\mathcal{E}_{\mu\nu}$ is constrained. In the vacuum case, one can easily check that the conservation equation is fulfilled. Our solution belongs to the case where the high-energy term $\mathcal{S}_{\mu\nu} \sim (T_{\mu\nu}^{(\omega)})^2$ is neglected [28]. It will play a role when one approaches the Planck scale. In some sense, one can state that this vacuum solution, which can analytically continue to Euclidean space, represents an instanton. As ´t Hooft stated [29], "The creation of a small black hole could be caused by an intervening gravitational instantons using the antipodal identification. Such instanton should be topological stable for all deformations barring scale transformations". We will return to this issue in Section 2.4.

Finally, some notes must be made about the physical interpretation of the dilaton $\omega$. We already mentioned that $\omega$ defines the scales and is observer-dependent. Locally, it is unobservable (comparable with a local gauge), particularly when considering a black hole horizon. It is fixed only if one knows the global spacetime and after choosing a coordinate frame with its vacuum state.

The "un-physical" spacetime $\tilde{g}_{\mu\nu}$ must obey equations of its own and should describe some of the physical phenomena that are taking place. Note that we still have the freedom of Equation (2), so we can adjust $\omega$ such that, for example, no naked singularities emerge, or at least occur at time infinity. Further, we still have the dilaton Equation (13). It turns out that this equation is superfluous, as expected. This can be seen when we consider $g_{\mu\nu} = \omega^2 \tilde{g}_{\mu\nu}$ as a transformation. Then, the Ricci scalar transforms as $\tilde{R} = \frac{1}{\omega^2} \left( R - \frac{6}{\omega} \nabla^{\mu} \nabla_{\mu} \omega \right)$. If $\tilde{R} = 0$ in a region where $\tilde{g}_{\mu\nu}$ is conformally flat, we obtain the dilaton equation for $V = 0$. If one Fourier transforms $\omega(r,t)$ to $\omega(k)$, then the wave vector $k$, with $k^2 = 0$ and pointing in the $+$ direction, determines the non-interacting massless particles. This follows from the $(+,+)$ component of the Ricci tensor. Because $\omega$ belongs to the gravitation sector, one could say that in the case of an evaporating black hole, these particles can be seen as Hawking radiation. So, the complementarity transformation that modifies the values of $\omega$, while keeping Equation (13) valid, switches on and off the effects these particles have on $\tilde{g}_{\mu\nu}$. An ingoing observer moving in the direction of the horizon will experience a different surrounding spacetime than the outside observer, and they disagree about the backreaction of the Hawking radiation. Therefore, they will also have different ideas about the vacuum state and the expectation value of $\omega^2$. Since $\tilde{R}_{\mu\nu}$ is not invariant under conformal transformation, the observers also disagree about the matter distribution.

One would eventually handle the dilaton field as a quantum field when approaching the Planck scale, while one could treat $\tilde{g}_{\mu\nu}$ (renormalized) classically[5].

Without the 5D contribution in the effective 4D Einstein equations, one obtains the solutions for $N^2$ and $\omega$ (compare with Equation (23))

$$N^2 = \frac{1}{4r^2} \frac{(6d_2^2 r^2 + 8d_2 r^3 + 3r^4 + D_1)}{D_2(t + d_3)^2 + D_3}, \quad \omega = \frac{1}{(r + d_2)t + rd_3 + d_2 d_3},$$ (28)

Again, one can write the r-dependent part as

$$N_1(r)^2 = \frac{3}{r^2} \int r(r + d_2)^2 dr.$$ (29)

So, the singularities are determined by a quartic equation, and the residue is determined by a quadratic equation.

## 2.2. Relation with the (2 + 1)-Dimensional Baňadoz–Teitelboim–Zanelli Solution

Gravity in a $(2+1)$-dimensional context has been recognized as a framework for studying GRT in relation to quantum-gravity issues. It can serve as a stepping stone toward understanding full four-dimensional spacetime. It also plays a prominent role

in understanding the AdS/CFT correspondence. In 1992, Baňados–Teitelboim–Zanelli (BTZ) [31] found an exact solution for a spinning black hole, which is axisymmetric and stationary. The black hole arises from identifications of points in $AdS_3$ spacetime by a subgroup of $SO(2,2)$. The action is

$$S = \int \sqrt{-g}\left(R + \frac{2}{l^2}\right)d^2xdt \tag{30}$$

with $\Lambda = -\frac{1}{l^2}$. The solution becomes

$$N_1^2 = \frac{\frac{r^4}{l^2} - 8MGr^2 + 16G^2J^2}{r^2} = \frac{4}{l^2r^2}\int r(r + l\sqrt{4GM})(r - l\sqrt{4GM}),$$

$$N^\varphi = -\frac{4GJ}{r^2}. \tag{31}$$

where $M$ is the mass, $J$ is the angular momentum, and $l$ is the scale where the curvature sets in. The spacetime is locally Minkowski [32]. If we compare this expression with our solution for Equation (23), we can conclude that the constant $a_2$ will be related to the mass and thus to the location of the horizon. The relation with the dynamical "uplifted" BTZ solution[6] was presented by Slagter [33]. It turns out that the cosmological constant must then be taken as zero.

It is remarkable that one can write the equation for the $r$-dependent metric component $N(r)$ for a general $k$ as a first-order differential equation

$$N_1^2 + rN_1\frac{\partial N_1}{\partial r} = b(r + a_2)^k, \tag{32}$$

where $b$ is a constant and $k \in \mathbb{Z}$. The general vacuum solution in the conformal model becomes

$$N_1(r) = \sqrt{\frac{1}{r^2}\left(\frac{(r + a_2)^{k+1}(r(k + 1) - a_2)}{k + 2} + \frac{a_2^{k+2}}{k + 2} + C\right)}. \tag{33}$$

It is clear that for the case $C = -\frac{a_2^{k+2}}{k+2}$, the zeros are for $r_H = -a_2$ with multiplicity $(k + 1)$ and for $r_H = \frac{a_2}{k+1}$[7]. For $k = 3$, we recover our 5D solution, and for $k = 2$, the BTZ solution. For a general $k$, that is, in a $k + 1$-dimensional warped spacetime, there will still be one real zero for positive r, while the other zeros are located at negative r, with multiplicity $(k + 1)$. For $C \neq 0$, there is only one real solution, whereas the other ones are complex.

### 2.3. Penrose Diagram

If one defines the coordinates,

$$dr^* \equiv \frac{1}{N_1(r)^2}dr, \qquad dt^* \equiv N_2(t)^2dt, \tag{34}$$

then our induced effective spacetime can be written as [25]

$$ds^2 = \omega^{4/3}\bar{\omega}^2\left[\frac{N_1^2}{N_2^2}\left(-dt^{*2} + dr^{*2}\right) + dz^2 + r^2(d\varphi + \frac{N^\varphi}{N_2^2}dt^*)^2\right], \tag{35}$$

due to the fact that the solution for the metric component $N$ is a quotient $\frac{N_1(r)}{N_2(t)}$. Further,

$$r^* = \frac{1}{4}\sum_{r_i^H}\frac{r_i^H\log(r - r_i^H)}{(r_i^H + b_2)^3}, \qquad t^* = \frac{1}{4C_2}\sum_{t_i^H}\frac{\log(t - t_i^H)}{(t_i^H + b_3)^3}. \tag{36}$$

The sum is taken over the roots of $(10b_2^3 r^2 + 20b_2^2 r^3 + 15b_2 r^4 + 4r^5 + C_1)$ and $C_2(t + b_3)^4 + C_3$, i. e., $r_i^H$ and $t_i^H$. This polynomial in $r$ defining the roots of $N_1^2$ is a quintic equation, which has an interesting connection with Klein's icosahedron solution. Further, one can define the azimuthal angular coordinate $d\varphi^* \equiv (d\varphi + \frac{N^\varphi}{N_2^2} dt^*)$, which can be used when an incoming null geodesic falls into the event horizon. $\varphi^*$ is the azimuthal angle in a coordinate system rotating about the z-axis relative to the Boyer–Lindquist coordinates. Next, we define the coordinates [34] (in the case of $C_1 = C_3 = 0$ and one horizon, for the time being)

$$
\begin{array}{lll}
U_+ = e^{\kappa(r^* - t^*)}, & V_+ = e^{\kappa(r^* + t^*)} & r > r_H \\
U_- = -e^{\kappa(r^* - t^*)}, & V_- = -e^{\kappa(r^* + t^*)} & r < r_H,
\end{array}
\tag{37}
$$

where $\kappa$ is a constant. The spacetime becomes

$$
ds^2 = \omega^{4/3} \bar{\omega}^2 \left[ \frac{N_1^2}{N_2^2} \frac{dU dV}{\kappa^2 UV} + dz^2 + r^2 d\varphi^{*2} \right].
\tag{38}
$$

If we compactify the coordinates,

$$
\tilde{U} = \tanh U, \qquad \tilde{V} = \tanh V,
\tag{39}
$$

then the spacetime can be written as

$$
ds^2 = \omega^{4/3} \bar{\omega}^2 \left[ H(\tilde{U}, \tilde{V}) d\tilde{U} d\tilde{V} + dz^2 + r^2 d\varphi^{*2} \right],
\tag{40}
$$

with

$$
H = \frac{N_1^2}{N_2^2} \frac{1}{\kappa^2 arctanh\tilde{U} arctanh\tilde{V} (1 - \tilde{U}^2)(1 - \tilde{V}^2)}.
\tag{41}
$$

We can express $r$ and $t$ as

$$
r = r_H + \left( arctanh\tilde{U} arctanh\tilde{V} \right)^{\frac{1}{2\kappa\alpha}}, \quad t = t_H + \left( \frac{arctanh\tilde{V}}{arctanh\tilde{U}} \right)^{\frac{1}{2\kappa\beta}},
\tag{42}
$$

with

$$
\alpha = \frac{r_H}{4(r_H + b_2)^3}, \qquad \beta = \frac{1}{4C_2(t_H + b_3)^3}.
\tag{43}
$$

Further, $N_1$ and $N_2$ are now functions in $(U, V)$ (or $(\tilde{U}, \tilde{V})$). In Figure 1, we illustrate the Penrose diagram. For $U = 0$ and $V = 0$, one obtains the horizons $r_H$ and $t_H$, as expected. The scale term $H$ is consistent with the features of the Penrose diagram. Remember that $ds^2$ and $H$ are invariant under the transformation $\tilde{U} \to -\tilde{U}$ and $\tilde{V} \to -\tilde{V}$. $\tilde{g}_{\mu\nu}$ is regular everywhere and conformally flat.

The hyperbolas in the Penrose diagram are given by

$$
UV = e^{2\kappa r^*} = (r - r_H)^{\frac{\kappa r_H}{2(r_H + b_2)^3}},
\tag{44}
$$

where $b_2$ contains the mass and $\kappa$ is a constant. So, when the black hole is evaporating, the hyperbola moves to the left in region I in the Penrose diagram. If $r$ is close to $r_H$, the hyperbola approaches the Planckian area. The antipodal points $P(X)$ and $P(\bar{X})$ are physically identified. In light cone coordinates, $u = r - t$ and $v = r + t$, one obtains a set of PDEs

$$
\Delta N = \frac{1}{1 + N^4} \mathcal{A}, \quad \Delta\omega = \frac{1}{1 - N^4} \mathcal{B}, \quad \Delta X = \frac{1}{1 - N^4} \mathcal{C},
\tag{45}
$$

where $\mathcal{A}, \mathcal{B}$, and $\mathcal{C}$ are expressions in $N, \omega$, and $X$, and their derivatives. Further, $\Delta \equiv \partial_u^2 + \partial_v^2$. These PDEs can be used in a numerical setting to follow the Hawking particles in the $(U, V)$ plane[8]. One can also prove that Laplace equations make sense on Klein surfaces [14].

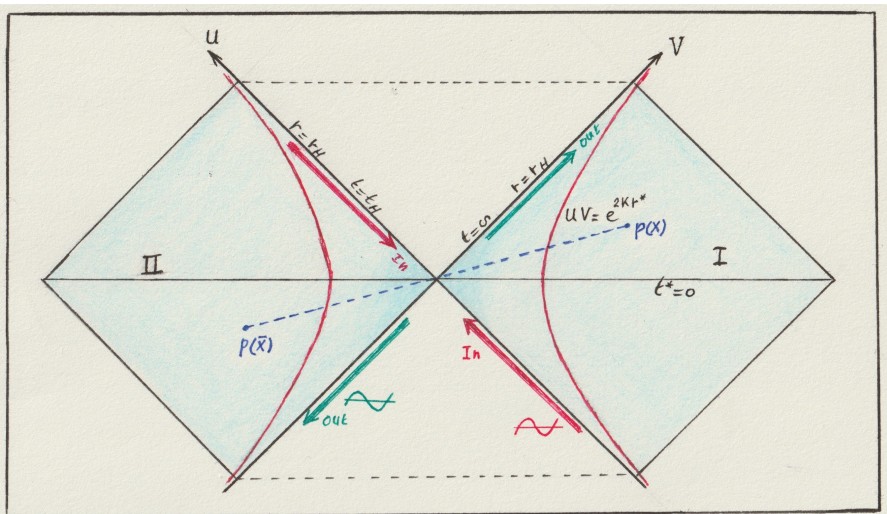

**Figure 1.** Penrose diagram. The antipodal points $P(X)$ and $P(\bar{X})$ are identified. Particles going in will generate waves that approach the horizon from the outside. Those passing through the horizon will reappear from "the other side" of the black hole. Note that $t^* = log(U/V) \sim log(t - t_H)$. In this approach the regions I and II are CPT invariant, implying that time runs backward in region II.

*2.4. Instanton Solution*

Let us write our (4 + 1)-dimensional spacetime on the five-dimensional Riemannian manifold:

$$ds^2 = \omega(\tau, r, y)^{4/3}\left[N(\tau, r)^2 d\tau^2 + \frac{1}{N(\tau, r)^2}dr^2 + dz^2 + r^2(d\varphi + N^\varphi(\tau, r)d\tau)^2 + dy_5^2\right]. \quad (46)$$

One easily obtains the field equations ($d = 4, 5$)

$$\ddot{\omega} = N^4\omega'' - \frac{d}{\omega(d-2)}(N^4\omega'^2 - \dot{\omega}^2), \quad (47)$$

$$\ddot{N} = \frac{3\dot{N}^2}{N} + N^4(N'' + \frac{3N'}{r} + \frac{N'^2}{N})$$
$$+ \frac{(d-1)}{(d-3)\omega}\left[N^5\left(\omega'' + \frac{\omega'}{r} + \frac{d}{2-d}\frac{\omega'^2}{\omega}\right) + N^4(\omega'N' - \dot{\omega}\dot{N}\right], \quad (48)$$

where the dot now represents $\partial_\tau$. Note that here, we are now dealing with hyperbolic PDEs instead of elliptic ones on the pseudo-Riemannian spacetime. The exact solution is again given by Equation (23) on the pseudo-Riemannian spacetime)[9]. Note that our original pseudo-Riemannian 5D spacetime delivered an effective 4D Riemannian spacetime on the brane, i.e., Equation (35). Now, we will have

$$ds^2 = \omega^{4/3}\bar{\omega}^2\left[\frac{N_1^2}{N_2^2}(d\tau^{*2} + dr^{*2}) + dz^2 + r^2 d\varphi^{*2}\right], \quad (49)$$

where $r$ can be expressed as $r^*$. The topology is $\mathbb{R}_+^2 \times \mathbb{R}^1 \times S^{1}$[10]. The $\varphi^*$ causes no problems because the $N^\varphi$ component decouples from the equations for $N$ and $\omega$. $\mathbb{R}_+^2$ is still conformally flat.

Instanton solutions play a fundamental role in field theory and possibly also in gravity theory. It is well-known that on Minkowski spacetime $\mathbb{R}^{d+1}$, the action of the static Yang–Mills–Higgs configuration $(\mathbf{A}, \Phi)$ equals the Euclidean action on $\mathbb{R}^d$. The field equations of the Euclidean action, for example, in $\mathbb{R}^4$ in this model, allow soliton solutions, which are time-independent finite energy solutions for the variational equations for the action density on the $\mathbb{R}^5$ Minkowski spacetime. For the $d = 4$ pure Yang–Mills case, they are called instantons. It turns out that their "curvatures" are self-dual[11]. In the book by Jaffe and Taubes [35], one can find a comprehensive overview of these issues.

One can say that the transition from Minkowski to Euclidean spacetime is a Wick rotation. What is the advantage of this "simplification"? Well, it has a big advantage in quantum field theory. One can construct a Euclidean propagator, and the action is then positive definite and finite. The propagator can be decomposed into a complete set of eigenfunctions of the Hamiltonian. Afterward, one Wick rotates the solution back to Minkowski spacetime. One can also define a Euclidean vacuum, i.e., the zero point of the potential energy density. The field approaches the vacuum asymptotically. Other nontrivial solutions are then instantons. They are useful because they dominate the Euclidean path integrals and can help find the Euclidean propagator. An important principle is then valid: "A static soliton in d space dimensions is completely equivalent to an instanton in d spacetime dimensions". In the case of a double-well potential with a degenerate minimum, the instanton interpolates between the two classical vacua. This degeneracy persists if one quantizes the model! The interested reader should consult Chapter 5 of Felsager [23].

It has been known for a long time that Euclidean self-dual gravitational fields exist. They have a vanishing classical action and nontrivial topological invariants. In some sense, they resemble the Yang–Mills instantons [36]. The asymptotically local Euclidean self-dual metrics imply that the path integral possesses the same weight as the flat vacuum metric. These solutions likely play an important role in understanding quantum gravity. There exists a conjecture that states that for a positive action $S$ and a non-singular positive definite asymptotically locally Euclidean metric with $R = 0$, $S = 0$ if the curvature is self-dual. The instanton solution in the vacuum case could also be used to explain the formation of a Planck-size black hole or study the very final stage of an evaporating black hole, which also takes place on Planckian scales [29].

Let us now compare our instanton solution with the well-known self-dual axially symmetric Eguchi–Hanson instanton [36] solution, where the Riemannian spacetime is

$$ds^2 = F(\rho)^2 d\rho^2 + \rho^2 \left[\sigma_x^2 + \sigma_y^2 + G(\rho)^2 d\sigma_z^2\right] \tag{50}$$

with $\rho^2 = \tau^2 + x^2 + y^2 + z^2$. Using the Newmann–Penrose formalism and imposing anti-self-duality on the spin connections, one obtains

$$F = \frac{1}{G}, \qquad G + \rho \frac{\partial_\rho G}{\partial \rho} = \frac{2 - G^2}{G} \tag{51}$$

This first-order differential equation is comparable with our first-order equation for $N$. One can easily check that the solution becomes

$$G = \sqrt{1 - \frac{a^4}{\rho^4}} \tag{52}$$

It satisfies Einstein's equations with anti-self-dual curvature. If one defines $u^2 = \rho^2\left(1 - \frac{a^4}{\rho^4}\right)$, then near the singularity $r = a$, i.e., $u = 0$, one recovers the two-sphere in polar coordinates for fixed $(\theta, \xi)$[12], $ds_2^2 = du^2 + u^2 d\varphi^2$. So, for $\rho \to a$, the spacetime is locally $\sim \mathbb{R}^2 \times S^2$, provided $\varphi \in [0, 2\pi]$[13]. Then, the Eguchi–Hanson spacetime is regular everywhere, geodesically complete, and singular-free. For $\tau \to 0$, $\mathbb{R}^2$ shrinks to a point on $S^2$. The manifold is homotopic to $S^2$ and has a Euler characteristic of 2. For large $\rho$, the metric approaches a

flat spacetime. The constant $\rho$ hypersurfaces are not three-spheres but three-spheres with antipodal points identified. The boundary is thus $SO(3) = P_3(\mathbb{R})$, for which $S^3 = SU(2)$ is the double cover. The metric is asymptotically locally Euclidean, $S^3/\mathbb{Z}_2$, but not globally. The entire manifold can be represented as a cotangent bundle of the complex plane $S^2 \sim P(\mathbb{C}^1)$.

We introduce once again a different radial coordinate $\bar{u}^4 = \rho^4 - a^4$ and the complex coordinates $\mathcal{V} = x + iy$, $\mathcal{W} = z + i\tau$ ($\bar{u}^2 = \mathcal{V}\bar{\mathcal{V}} + \mathcal{W}\bar{\mathcal{W}}$). The spacetime then becomes, on $\mathbb{C}^2/\{0\}$ (Kähler form),

$$ds^2 = \frac{\bar{u}^2}{\sqrt{\bar{u}^4 + a^4}}(d\mathcal{V}d\bar{\mathcal{V}} + d\mathcal{W}d\bar{\mathcal{W}}) + \frac{a^4}{\sqrt{\bar{u}^4 + a^4}}\partial\bar{\partial}\log(\bar{u}^2) \tag{53}$$

Again, we see that $\bar{u} = 0$ ($\rho = a$) causes problems, but this apparent singularity can be removed by the *antipodal identification* $(\mathcal{V}, \mathcal{W}) \sim (-\mathcal{V}, -\mathcal{W})$. At infinity, the coordinate $x_\mu$ is identified with its time-parity conjugate $-x_\mu$. The spacetime originates at $\rho = a$ and represents an instanton-like bump in the curvature. In our case, we must consider, for example, the vacuum case, where the $\tilde{g}_{\mu\nu}$ is conformally flat, and also consider $T_{\mu\nu}^{(\omega)}$. One can easily check that $\tilde{R} = 0$ for our "un-physical" spacetime. In order to prove that our metric is self-dual, one needs the Cartan structure components. The advantage is that one has to solve first-order differential equations. Note that our solution has a Eguchi–Hanson-type form

$$ds^2 = N^2(r, \tau)d\tau^2 + \frac{1}{N^2(r, \tau)}dr^2 + r^2(d\sigma_x^2 + d\sigma_y^2) \tag{54}$$

where the metric component $N^2$ is now a fifth-order polynomial instead of a fourth-order polynomial in $r$. Moreover, our solution is $(r, \tau)$-dependent.

## 3. The Non-Vacuum Case

We take now for the gauge potential $A_\varphi = -\frac{n}{e}$, i.e., $P(t, r) = 0$ in Equation (5), and a potential $V(\Phi, \omega)$. The equations then become

$$\ddot{N} = -N^4\left(N'' + 3\frac{N'}{r} + \frac{N'^2}{N}\right) + 3\frac{\dot{N}^2}{N} + \frac{2N}{(\eta^2X^2 + \omega^2)}\left[y_0 N^2 V\right.$$
$$+ 3N^4(\eta^2X'^2 + \omega'^2) - 3(\eta^2\dot{X}^2 + \dot{\omega}^2) - \frac{3}{4r}N^4(\eta^2XX' + \omega\omega')\bigg], \tag{55}$$

$$\ddot{X} = N^4\left(X'' + \frac{X'}{r} + 2\frac{X'N'}{N}\right) + 2\frac{\dot{X}\dot{N}}{N} + \frac{N^2}{\eta}\frac{\partial V}{\partial X}$$
$$+ \frac{2X}{(\eta^2X^2 + \omega^2)}\left[N^4(\eta^2X'^2 + \omega'^2) - (\eta^2\dot{X}^2 + \dot{\omega}^2) + \frac{y_0}{3}N^2V\right], \tag{56}$$

$$\ddot{\omega} = N^4\left(\omega'' + \frac{\omega'}{r} + 2\frac{\omega'N'}{N}\right) + 2\frac{\dot{\omega}\dot{N}}{N} + y_0 N^2\frac{\partial V}{\partial \omega}$$
$$+ \frac{2\omega}{(\eta^2X^2 + \omega^2)}\left[N^4(\eta^2X'^2 + \omega'^2) - (\eta^2\dot{X}^2 + \dot{\omega}^2) + \frac{y_0}{3}N^2V\right]. \tag{57}$$

From the superfluous $\omega$ equation, we obtain that the potential must be taken as

$$V(X, \omega) = \beta_1 X^{\beta_2}\omega^{\frac{2}{3} - \frac{\eta\beta_2}{|y_0|}}, \qquad (\beta_1, \beta_2)cst. \tag{58}$$

which is consistent with the tracelessness of the energy-momentum tensor[14]. Observe that the scale of the extra dimension enters the equations through $y_0$. It is remarkable that we can obtain a quartic conformal invariant matter coupling in the Lagrangian

$$M \sim X^2\omega^2, \tag{59}$$

for a suitable combination of the parameters, i.e.,

$$|y_0| = \frac{3}{2}\eta. \tag{60}$$

If we inspect the equations for the scalar and dilaton fields in the case of a flat spacetime $N = 1$, we obtain the two potentials

$$V_X = \frac{\alpha\left(\eta X^2(3\eta\beta + 2y_0) + 3\beta\omega^2\right)}{3\eta(\eta^2 X^2 + \omega^2)} X^{\beta-1}\omega^{\frac{(2y_0 - 3\beta\eta)}{3y_0}}, \tag{61}$$

$$V_\omega = \frac{\alpha\left(\eta^2 X^2(2y_0 - 3\eta\beta) + \omega^2(4y_0 - 3\beta\eta)\right)}{3(\eta^2 X^2 + \omega^2)} X^\beta \omega^{-\frac{(y_0 + 3\beta\eta)}{3y_0}}. \tag{62}$$

In this special case, one can still integrate the functional integral over $\omega$ exactly. So, non-conformal matter does not affect the conformal invariance of the effective action after integrating over $\omega$ [18,21]. One can conjecture that in the non-vacuum case, this still holds.

The equations for $N^\varphi$ are

$$\dot{N}^{\varphi\prime} = -3N^{\varphi\prime}\frac{(\eta^2 X\dot{X} + \omega\dot{\omega})}{(\eta^2 X^2 + \omega^2)},$$

$$N^{\varphi\prime\prime} = -3N^{\varphi\prime}\left[\frac{1}{r} + \frac{(\eta^2 XX' + \omega\omega')}{(\eta^2 X^2 + \omega^2)}\right], \tag{63}$$

and the general solution is

$$N^\varphi = f\int\frac{1}{r^3(\eta^2 X^2 + \omega^2)^{3/2}}dr, \tag{64}$$

where $f$ is a constant. The scalar and dilaton equations differ only by the potential term. This difference is evident in the numerical solution shown in Figures 2 and 3. Note that $n$ has disappeared from the PDEs. We have incorporated the winding number (or vortex number) $n$ into the constant gauge field $|A_\varphi| = \frac{n}{e}$[15]. It is also possible to isolate an equation for $\ddot{X}$ and $\ddot{\omega}$ from the Einstein equations, i.e.,

$$\omega\ddot{\omega} + N^4\left(\omega\omega'' - 2\omega'^2\right) - 2\dot{\omega}^2 = X\ddot{X} + N^4\left(XX'' - 2X'^2\right) - 2\dot{X}^2. \tag{65}$$

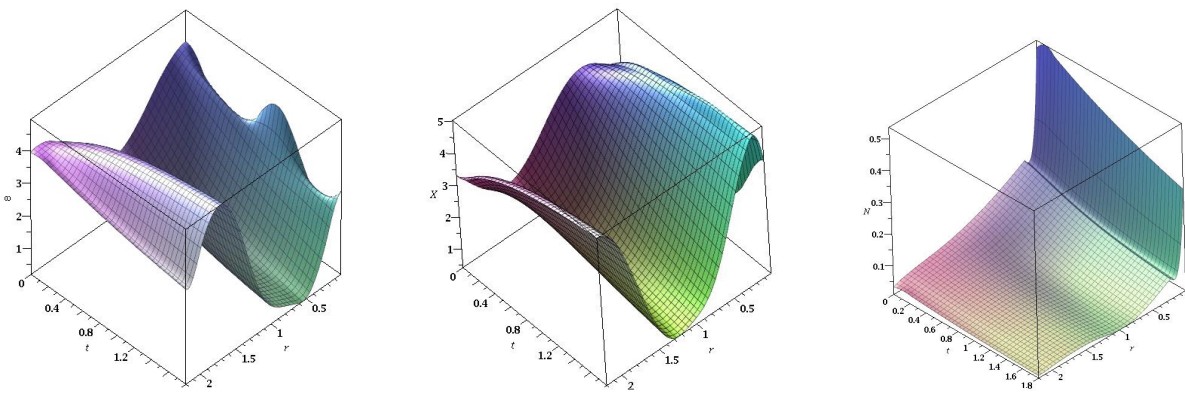

**Figure 2.** Stable solutions of $X, \omega$, and $N$ for the case of a constant gauge field. The potential is $V = \beta_1 X^{\beta_2}\omega^{\frac{2}{3} - \frac{\eta\beta_2}{y_0}}$. We used a cosine function for the initial values of the scalar field and a sine function for the dilaton field. For $N$, we used the vacuum solutions. Further, we applied Neumann boundary conditions. Note that $N$ develops a singularity and approaches a constant value with increasing time. The solution critically depends on the parameters of the potential, for example, the scale $y_0$ of the extra dimension.

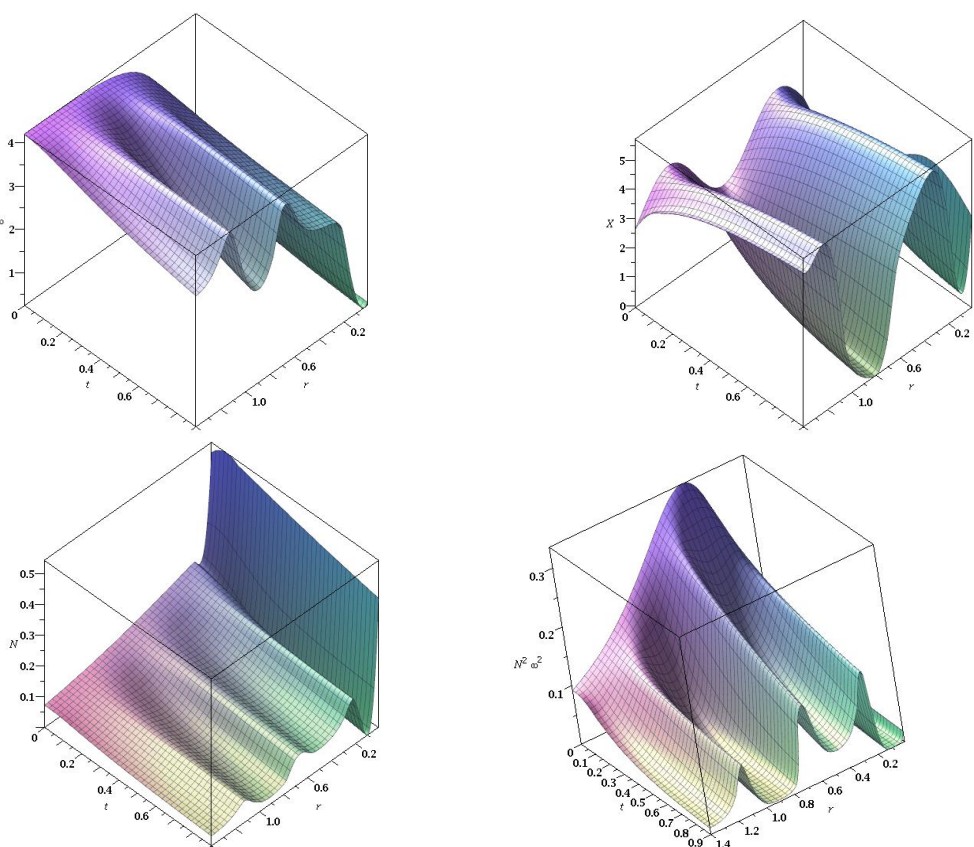

**Figure 3.** Same as in Figure 2, but now we use a tanh function as the initial value for $\omega$. We observe that the polynomial behavior in the metric component $N$ is induced by the dilaton. We also plotted the total metric component $\omega^4 N^2$.

Thereafter, we obtain

$$\ddot{\omega} + N^4\left(\omega'' - \frac{2\omega'^2}{\omega}\right) - \frac{2\dot{\omega}^2}{\omega} = \frac{C}{\omega}, \tag{66}$$

and the same equation for $X$. For $C = 0$, we recover the equation for $\omega$ in the vacuum case. The case of $C \neq 0$ is more interesting. Again, the exact solutions for $\omega$, $X$, and $N$ can be found in terms of elliptic functions. The potential $V$ appears in the constraint equation for $\dot{N}$. An exact solution for $\omega$ becomes

$$\omega = \frac{1}{(r - r_H)JacobiSN(\frac{1}{2}\sqrt{2C}(r - r_H)t, I)} \tag{67}$$

One might wonder if an estimate for the value of the extra dimension $y_0$ can be obtained. In the following sections, we will try to find a possible method using the elapsed proper time of a particle when it moves in the bulk space and back to the brane[16].

Our horizon, $r_H$, is determined by $a_i$ (see Equation (23)). We assume that $y_0 \sim 10^{-5}$ m [28] and let us take $r_H \sim 10^7$ m (see, for example, Maldacena [38][17]). Using the wormhole construction, he obtained an elapsed time of the order $10^{-2}$ s, while $t$ would be huge, of the order of tens of thousands of years for somebody looking from the outside.

The proper traversal time, $t_e$, of the Klein surface will be of the order of $r_H$. Further, the trajectory on the Klein surface in the bulk, $l_b$, will be of the order $\frac{t}{t_e}$, where $t$ is the asymptotic time and $t_e$ is the proper time. It would be of interest if one could measure this elapsed time in Hawking radiation. One could obtain an estimate of $y_0$.

## 4. Application of the Antipodal Boundary Condition

*4.1. The Black Hole Paradoxes*

It is well-known that for smaller scales, the spacetime in Equation (6) results in quantum-gravitational effects. In the vicinity of the horizon, pair production will take place due to the high curvature. It was Hawking's great achievement to calculate the radiation produced during the evaporation of a black hole [1]. He found that black holes radiate as black bodies with a thermal spectrum. So, the black hole would be in a mixed state. This radiation is random with a probability distribution controlled solely by the parameters of the black hole, i.e., the mass, angular momentum, and charge. However, the collapse started out of a pure state.

Since different states can have the same mass and angular momentum, this means that many initial physical states evolve into the same final stage. So, information on the details of the initial stage is lost. When one particle of the Hawking pair falls into the black hole and the other one escapes, they are in a pure quantum mechanical state, and the wave functions satisfy the Schrödinger equation. Pure quantum states are characterized by maximal information, i.e., zero entropy, so one has a conflict because pure states seem to evolve in mixed quantum states. On the other hand, the laws of quantum mechanics are unitary and thus time-reversible. The information is preserved. The lost information must reappear somehow. Over the past few decades, many physicists have proposed different approaches to tackle this epic problem. A possible solution was addressed by Almheri et al. [2]. They suggested that there is a firewall just outside the horizon, with a thickness of the order of the Planck length. The entanglement will be broken between the infalling and outgoing particles. This would release a large amount of energy, creating a firewall. An observer just inside the horizon would see the trapped information, while an observer just outside would find the information "scrambled" beyond recognition. So, there is a "complementarity" problem, which is not disastrous because they cannot exchange information. Matter falling through the horizon would immediately be burned at this firewall. It could be possible that in the future, one could eventually detect this "bounce" against the firewall in the gravitational wave spectrum. Nevertheless, it conflicts with general relativity. One can now conjecture that three statements cannot be simultaneously true:

1. Hawking radiation is in a pure state.
2. The information is emitted from the stretched horizon. One can use GRT to describe this phenomenon.
3. An infalling observer encounters nothing special when crossing the horizon and is burned at the firewall.

Could one solve the paradoxes by giving up Einstein's equivalence principle, or unitarity? One could even suggest that quantum mechanics must be revised [39].

Maldacena et al. [3] proposed the "ER = EPR" hypothesis, which is based on the suggestion that connectivity in spacetime is equivalent to quantum entanglement and is related to locality. The two regions in the Penrose diagram of the maximally extended Schwartzschild solution would connect two black holes connected by an Einstein–Rosen bridge (wormhole). One proposes that the entangled Hawking particles remain connected by the wormhole, which connects the pair of "entangled" black holes. This conjecture is an extrapolation of the observation that a maximally extended AdS–Schwarzschild black hole with a wormhole connection is dual to a pair of maximally entangled thermal conformal field theories via AdS/CFT correspondence. In this model, a firewall is not necessary. A great disadvantage is the fact that the two black holes in regions I and II in the Penrose diagram will feel their gravitation field [40]. Further, this model would suggest that any entangled pair of states are connected by Planck-scale wormholes. If one detects these mini wormholes at CERN, then one has a strong argument. Quite recently, Maldacena et al. [38] used this model to calculate the traversal time of the Einstein–Rosen bridge. They considered the brane-world model of Randall–Sundrum (RS) [11,12], with one large extra dimension. Here, we will apply a slightly different approach to the RS model.

*4.2. Antipodicity*

In a former study [25,26], we considered the black hole paradoxes in the context of a warped 5D RS spacetime. Further, we imposed the antipodal boundary condition on a Klein bottle surface.

In general, a spacetime with a given local geometry admits different possible global isometries. One can consider the modification of the spacetime topology in the form $\widehat{\mathcal{M}}/\Gamma$, where $\Gamma$ is a discrete subgroup of isometries of $\mathcal{M}$ without fixed points. $\widehat{\mathcal{M}}$ is non-singular and is obtained from its universal covering $\mathcal{M}$ by identifying points equivalent under $\Gamma$. A particularly interesting case is obtained when $\Gamma$ is the antipodal transformation on $\mathcal{M}$

$$J: \quad P(X) \rightarrow \widehat{P}(\widehat{X}). \tag{68}$$

where the light cone of the antipode of $P(X)$ intersects the light cone of $\hat{P}(\hat{X})$ only in two points (at the boundary of the spacetime). At the intersection, one then identifies the antipodal points. This idea has already been mentioned by Schrödinger(1957) [9]. His so-called "elliptic interpretation" was historically first studied in cosmology, i.e., in the de Sitter spacetime. This method was later extended by Sanchez et al. [41] and Gibbons [42]. Quite recently, 't Hooft [10,40] revised this approach.

Some physicists are convinced that the black hole paradoxes could be solved by changing the topology around the horizon. A fundamental issue that is omitted in all the treatments of the effects near the horizon of a black hole is the time dependency of the spacetime structure near the horizon. The emitted Hawking particle will have a backreaction effect on the spacetime. The antipodal boundary condition could be an alternative route. As already mentioned, the issue of the entangled ingoing and outgoing Hawking particles causes problems. The information of the ingoing particles must get back somehow.

The very basic idea behind the identification is the identification of the two regions (I and II) in the Penrose diagram (see Figure 1). Region II is the antipode of I. Consequently, there is no hidden sector. It could restore quantum purity and CPT invariance, properties usually not found in thermodynamic objects as Hawking proposed for a black hole. The approach of 't Hooft relies on the unitarity of the S-matrix and the non-orientable structure of the hypersurface (a Möbius surface). Hawking radiation is only locally thermal, but globally, quantum states evolve unitarily, assuming the entanglement of the Hawking particles. The trick is that the infalling particle returns to the antipode.

In order to maintain the Hartle–Hawking vacuum state as a pure state instead of a thermodynamically mixed one for a faraway observer, one applies the antipodicity on the spherical coordinates $(\theta, \varphi)$ [10,17][18]. The result is that the Hawking particle remains maximally entangled and no firewall is necessary. The antipodal boundary condition[19] minimizes the number of unconventional assumptions or the need to rely on full string theory or "fuzzyballs". It will require that the local laws of physics are invariant across the black hole boundary [29]. Usually, one considers the Hawking particles as excitations of low-energy particles that fulfill the standard model and apply perturbative quantum gravity on a fixed background metric (i.e., the gravitons). The distant observer notices the Hilbert space of low-energy particles at a given time. One then maps the states from later times to earlier times, one on one. One essentially bypasses the Planck area. It manifests itself only through the higher modes of the spherical harmonics (in the angle variables). A unitary S matrix can then be constructed if one applies a cutoff for the higher modes of the harmonics.

The antipodal boundary condition consists of, in Kruskal–Szekeres coordinates, the map: $I \rightarrow II : (U, V, \theta, \varphi) \rightarrow (-U, -V, \pi - \theta, \varphi + \pi)$. The firewall could then be solved, i.e., the ingoing particles transform into outgoing particles. One then uses the quantum mechanical approximation model of the partial wave scattering of excited harmonic oscillators. The desired unitary S-matrix is then obtained. The spherical harmonics satisfy $Y_{l,m}(\pi - \theta, \pi + \varphi) = (-1)^l \mathbf{Y}_{l,m}(\theta, \varphi)$, so only odd modes contribute. One assumes that the

separation of the coordinates $(\theta, \varphi)$ is possible. In our case (see Section 5), we have the factor $(-1)^{n+m}$, where $n$ is the winding number and $\mathbf{Y}_m$ represents the Bessel functions. For $n = m - 1$, it is always negative.

The quantum mechanical principle can be summarized [43,44] as follows: ingoing wave packets are gravitationally backreacted, and they are strongest for the outside observer. Most of the information then passes through the antipodal region, and a small fraction is reflected back. The ingoing position is imprinted on the outgoing momenta. Highly localized momenta of ingoing waves transform into two outgoing pieces, i.e., transmitted and reflected ones. The positions are then highly delocalized. Large-wavelength Hawking particles are produced out of short-wavelength waves. The entanglement would need a wormhole connection between regions I and II in the Penrose diagram or the antipodal boundary condition of 't Hooft's method. In our case, we do not need a wormhole construction: the antipodal identification is performed on the Klein surface. No reflection procedure is necessary. There is only a time delay between the outgoing Hawking particles and the ingoing particles traveling on the Klein surface and leaving at the antipode. The main task was how to describe the quantum mechanical effects on the time evolution of the Hawking particles falling into region I of the Penrose diagram. It is possible to describe these effects at low energies (as the Hawking particles are), such that initially, the backreaction on spacetime is negligible. One can "dynamically" handle the gravitational effects due to particles moving in or out of horizons.

The Planck scale comes into play when one goes to a high wave number $l$ of the spherical modes. The cutoff will then be determined by the number of micro-states in order to reproduce the Hawking radiation value.

How could the primordial black hole come into being and grow? It could be a gravitational instanton. In 't Hooft's model, one cuts out a small $S^3$ region in spacetime and applies the cut-and-paste method of the antipodal map. The topology is then $\mathbb{R} \times S^3/\mathbb{Z}^2$. In our model, we have the Klein surface embedded in 5D, with topology $\mathbb{R}^2_+ \times \mathbb{R} \times S^1$. In Section 2.4, we found this instanton solution on the 5D warped Riemannian manifold, which was equivalent to the solution on the pseudo-Riemannian manifold. Further, the solution of the effective 4D spacetime was the same apart from some constants. The dilaton solution, treated as a quantum field for the local observer, determines the scale for the outside observer. When more particles fall in, the black hole will grow until the evaporation takes the upper hand. The most complicated issue in antipodal mapping is the treatment of the gravitational interaction. See Section 5.4.

## 5. Antipodicity in the Conformal Dilaton–Higgs Model

### 5.1. Cylindrical Harmonics and the Stress Tensor

In our model, we can write the scalar as (compared to Equations (5))

$$\phi_{mn} = \eta X(r,t) e^{in\varphi} \mathbf{Y}_m(\varphi), \tag{69}$$

where $X(t,r)$ fulfills the partial differential equations in Section 2. In the case of a potential $V = \frac{A}{r^2}(X^2 + \omega^2)$, where A is a constant, the equation for $Y_m(\varphi)$ can be separated as

$$\frac{\partial^2}{\partial \varphi^2} \mathbf{Y}_m(\varphi) = -m^2 \mathbf{Y}_m(\varphi) - y_0, \tag{70}$$

where $y_0$ is still the scale of the extra dimension. The solution becomes

$$\mathbf{Y}_m = A e^{im\varphi} + B e^{-im\varphi} + \frac{y_0}{m^2}. \tag{71}$$

So, we have

$$\phi_{mn} = \eta X(r,t) \sum_{m,n} \left[ A_m e^{i(m+n)\varphi} + B_m e^{-i(m-n)\varphi} + \frac{y_0}{m^2} e^{in\varphi} \right]. \tag{72}$$

Here, we have two mode-expansion numbers $(m, n)$, where $n$ is the winding number. For a more general potential, the separation of variables is harder to perform. It is clear that we then need, for example, a high-frequency approximation [45,46]. When we approach the very small scale, we already remarked that the scalar field and the dilaton are equal, apart from the potential. Suppose we can write the solution of the scalar equation using creation and annihilation operators on a Cauchy surface $\Sigma$

$$\Phi = \sum_{\nu mn} \phi_{\nu mn} a_{\nu mn} + \phi_{\nu mn}^* a_{\nu mn}^\dagger, \tag{73}$$

where $\Phi_{\nu mn}$ is now $X_\nu(t, r)\mathbf{Y}_m(\varphi)e^{in\varphi}$, which depends on $\nu \in \mathbb{R}$. In Eddington–Finkelstein coordinates, one usually makes the distinction between the outgoing and ingoing waves using $(U, V)$ coordinates. In our situation, we can also keep $(r, t)$ because in region II of the Penrose diagram, t runs backward[20]. To obtain the stress-tensor operator components $\langle 0|\hat{T}_{ab}|0\rangle$, one needs the derivatives of $\hat{\phi}$

$$\partial_t \hat{\phi} = \int d\nu \sum_{m=-\infty}^{\infty} \frac{\nu}{2\pi} \left[ \partial_t X_\nu(r, t)\mathbf{Y}_m(\varphi)e^{in\varphi}\hat{a}_{\nu mn} + \partial_t X_\nu^*(r, t)\mathbf{Y}_m^*(\varphi)e^{-in\varphi}\hat{a}_{\nu mn}^\dagger \right]$$

$$\partial_r \hat{\phi} = \int d\nu \sum_{m=-\infty}^{\infty} \frac{\nu}{2\pi} \left[ \partial_r X_\nu(r, t)\mathbf{Y}_m(\varphi)e^{in\varphi}\hat{a}_{\nu mn} + \partial_r X_\nu^*(r, t)\mathbf{Y}_m^*(\varphi)e^{-in\varphi}\hat{a}_{\nu mn}^\dagger \right]$$

$$\partial_\varphi \hat{\phi} = \int d\nu \sum_{m=-\infty}^{\infty} \frac{\nu}{2\pi} \left[ X_\nu(r, t)\left( i(m+n)Ae^{i(m+n)\varphi} - i(m-n)Be^{-i(m-n)\varphi} \right)\hat{a}_{\nu mn} + \dots \right]. \tag{74}$$

Note that the integration over the mode space is performed with a factor of $\frac{k}{2\pi}$ to obtain the correct commutation relations for the creation and annihilation operators, i.e., delta functions. Next, we can calculate $\langle 0|\partial_a \hat{\phi}\partial_a \hat{\phi}|0\rangle$

$$\langle 0|\partial_t \hat{\phi}\partial_t \hat{\phi}|0\rangle] = 2 \int d\nu \sum_{m=-\infty}^{\infty} \sum_{n=odd} \frac{\nu}{2\pi} |\partial_t X_\nu|^2 \mathbf{Y}_m^2 e^{2in\varphi}$$

$$\langle 0|\partial_r \hat{\phi}\partial_r \hat{\phi}|0\rangle] = 2 \int d\nu \sum_{m=-\infty}^{\infty} \sum_{n=odd} \frac{\nu}{2\pi} |\partial_r X_\nu|^2 \mathbf{Y}_m^2 e^{2in\varphi}$$

$$\langle 0|\partial_\varphi \hat{\phi}\partial_\varphi \hat{\phi}|0\rangle] = 2 \int d\nu \sum_{m=-\infty}^{\infty} \sum_{n=odd} \frac{\nu}{2\pi} X_\nu^2 \left( |\partial_\varphi \mathbf{Y}_m|^2 + n^2 \mathbf{Y}_m^2 \right). \tag{75}$$

We can perform the same calculation for $\omega$. In principle, using this approach, we can obtain the desired $\langle 0|\hat{T}_{00}|0\rangle$ to make statements about the vacuum. However, we need expressions for $(x, \omega)$ and their first derivatives. Although the vacuum is related to the minimum of the Hamiltonian, i.e., the lowest energy state, it becomes troublesome in the non-flat spacetime. First, we are dealing with a time-dependent situation. Second, in treating the evaporation process, we have different notions for the local observer and the faraway observer. In our situation, we need both the scalar field and the dilaton $\omega$, and both are conformally invariant. So, we can use $\Omega$ (see Equation (2)). As we shall see, this will help us formulate the local vacuum. Because our model relies on a 5D warped spacetime, we extend the antipodicity to the Klein hypersurface in $\mathbb{R}^4$.

*5.2. Motivation for the Klein Bottle Surface*

In our model, we need the Klein surface $\mathbb{K}$, which is the compact sum of two projective planes. Our conjecture is that our solution can be seen as a dynamical solution embedded in our $(4 + 1)$-dimensional spacetime. More precisely, if $\mathcal{N}$ is the submanifold with the metric $g$, i.e., the metric in our effective 4D spacetime, then $\mathcal{N}$ is diffeomorphic to the hyperbolic 5D spacetime $\chi : \mathbb{R}^5 \to \mathcal{N}$. The Riemannian manifold $(\mathcal{N}, g)$ must be conformally flat, which is the case here. Physicists are now interested in the topology of moduli spaces of self-dual connections on vector bundles over Riemannian manifolds. One reason is that on these spaces, the instanton approximation to the Green functions of Euclidean quantum

gravity Yang–Mills theory can be expressed in terms of integrals over moduli spaces. One then needs the metric and volume form of the moduli spaces. From the investigations of Groisser et al. [47], we conjecture that we can consider $\mathbb{K}$ as a four-sphere in our hyperbolic pseudo-Riemannian spacetime. This is fueled by the solution in Section 4.2 and the work on the orientable counterpart model of Groisser (and the references therein). More precisely, on a Riemannian five-manifold, one can prove that there exists a coordinate diffeomorphism $\zeta : \mathbb{R}^5 \to \mathcal{N}$ for which the pullback of the metric $g$ on $\mathcal{N}$ is given by $(\zeta^* g)_{ij} = \Psi^2(\mathbf{x}) \delta_{ij}$. Further, the Riemannian manifold $(\mathcal{N}, g)$ is conformally flat, with a finite radius and volume. The action of $SO(5)$ on $S^4$ induces an isometry on $\mathcal{N}$ whose pullback, via $\zeta$, is the usual $SO(5)$ action on $\mathbb{R}^5$. So, $\mathcal{N}$ can isometrically be included as the interior of a compact Riemannian manifold, say $\bar{\mathcal{N}}$, whose boundary $\partial \bar{\mathcal{N}}$ is isometric to the four-sphere of constant radius. The embedding $\partial \bar{\mathcal{N}} \to \mathcal{N}$ is totally geodesic. The sphere $\partial \bar{\mathcal{N}}$ is conformally equivalent to the original manifold $(S^4, g)$, and points on $\partial \bar{\mathcal{N}}$ correspond to instantons that are concentrated at a single point on $S^4$. It is remarkable that $\Psi(\mathbf{x})$ is determined by a PDE that is comparable to our scalar equation ($\mathcal{N}$ has no constant curvature). A remark must be made about the $\mathbb{Z}_2$ symmetry in the original description of the antipodal mapping [29]. At the border of regions I and II in the Penrose diagram, the antipodes on a three-sphere are glued together, and the transverse $(\theta, \varphi)$ part is a projected two-sphere. In our model, it is replaced with the projected three-sphere $(z, \varphi, y_5)$ using the $\mathbb{Z}_2$ symmetry of the bulk space $(U, V, z, \varphi, y_5)$. No cut-and-paste procedure is necessary. There is another strong argument for considering the topology of the Klein surface. In short, we blow up the four-manifold to 5D to handle the singularities in the curvature and apply the antipodal map. One can mathematically formulate the topology of a four-manifold using self-dual connections over the Riemannian $S^4$ [48]. It depends only on the conformal class of the Riemannian metric. This self-dual connection can be interpreted by the conformal map $\mathbb{R}^4 \to S^4 / \{0\}$ as a self-dual connection or "instanton".

In our 5D RS model (with a finite number of singularities), we found that the metric is determined by $N$ and $\omega$ (see Section 2). The solution for $N$ in the effective 4D spacetime is the same, whereas the $\omega$ contribution is different. This is solely due to the contribution of $\mathcal{E}_{\mu\nu}$ (Equation (8)). If we switch to the Riemannian case ($t \to i\tau$, a Wick rotation), the solutions for both $N$ and $\omega$ remain unaltered. Moreover, $\tilde{g}_{\mu\nu}$ is conformally flat. The embedding of the Klein surface was performed using the extra dimension $y$, and the effective spacetime is conformally flat. Consider now the ball $B^4 \subset \mathbb{R}^4$ with the induced metric $\omega^{4/3} \bar{\omega}^2 \tilde{g}_{\mu\nu}$ (see Equation (40).[21]), so the scale is $\omega^4$. Then, the ball $\frac{r}{2\omega} B^4 \subset \mathbb{R}^4$ has a curvature $\leq \frac{const}{|\omega|^4}$. We interpret the Riemannian 5D warped spacetime (Equation (46)) as an open five-ball, as an instanton on $\mathbb{R}^4$. In the pseudo-Riemannian spacetime in Equation (6), the boundary is the non-orientable Klein surface, which we used for the antipodicity in order to maintain the pure states of the Hawking particles. The importance of the instanton trick is the fact that in the Riemannian space (or more easily, in Minkowski space), they play a crucial role in calculating path integrals [23]. It turns out that a static solution in $m$ space dimensions is completely equivalent to an instanton in $m$ spacetime dimensions. Now, remember that for the $S^2$, we could apply the stereographic projection $S^2 \to \mathbb{R}P^2 \to \mathbb{C}P^1$. To ensure a one-to-one mapping, the $\mathbb{Z}_2$ symmetry identification is applied: the two antipodal planes are identified, $z \to -\frac{1}{\bar{z}}$ (see Figure 4). In our situation, we uplift the projection to $S^3 \subset \mathbb{R}^4$ and use cylindrical coordinates $(r, \varphi)$.

Using the Hopf fibration, one then establishes the connection with real spacetime (for details, see Slagter [25] and Urbantke [49,50]). Another important characteristic of the non-orientable Klein surface is the fact that meromorphic functions remain constant [14]. Our solution for $N$ is meromorphic[22]. Further, the Klein surface is homeomorphic to the connected sum of two projective planes.

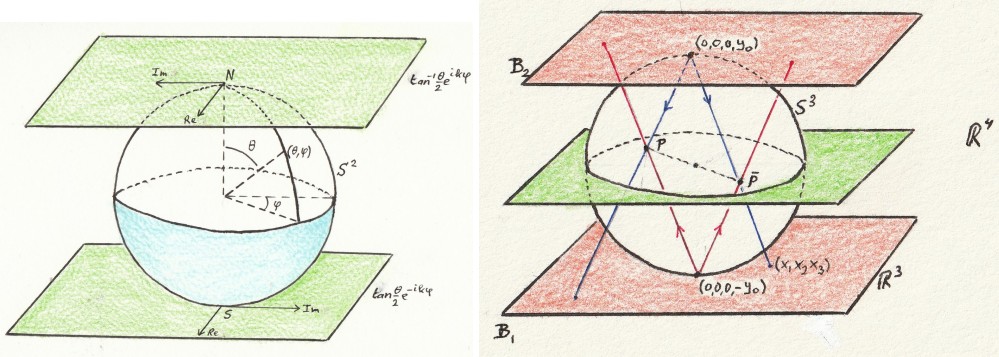

**Figure 4.** Left: Stereographic projection of $S^2 \to \mathbb{R}P^2 \to \mathbb{C}P^1$. To ensure a one-to-one mapping, the $\mathbb{Z}_2$ symmetry identification is applied: the two antipodal planes are identified, $z \to -\frac{1}{\bar{z}}$. Right: Stereographic projection of $S^3 \subset \mathbb{R}^4 \simeq \mathbb{C} \times \mathbb{C}$ also with antipodal identification, representing the projection of an embedded Klein bottle.

In the past, extensive research has been conducted on the fundamental group structure of compact surfaces in $m$ dimensions. It is clear that, for example, the torus possesses the structure of two infinite cyclic groups. For the projective plane, the cyclic group of order 2. For the Klein bottle, the fundamental group can also be presented [37]. We constructed our model in the 5D RS warped spacetime. So, the question is how we can imagine the evolution of the Hawking particles created near the horizon, say point $P(U, V, z, \varphi, y_5)$, as shown in Figure 4. The ingoing particle can travel on the Klein surface in order to leave in the antipode $\bar{P}(-U, -V, -z, \varphi + \pi, -y_5)$ on the horizon. $J : P \to \bar{P}$ is an inversion. The real physics still takes place on the brane. Effectively, we replaced the "cut-and-paste" method in $\mathbb{R}^3$ with transport in $\mathbb{R}^4$ (via the Hopf fibration of the three-sphere in $\mathbb{R}^4$).

We know that complete embedded surfaces in $\mathbb{R}^3$ must be orientable; otherwise, we have self-intersection, which changes the genus. Non-orientable surfaces immersed in $\mathbb{R}^3$ do not have global Gauss maps and thus do not have a well-defined mean curvature. So, the application of the Klein surface embedded in $\mathbb{R}^4$ is quite interesting in our context. We found that the system of PDEs can be solved on a conformally flat Riemannian spacetime. Therefore, the spacetime emerges from an instanton. First, we complexify our hypersurface [25]. Let us write our coordinates $(r, z, y_5, \varphi^*)$ as[23]

$$\mathcal{V} = z + iy_5 = Re^{i\varphi_1}, \qquad \mathcal{W} = x + iy = re^{i\varphi_2}, \tag{76}$$

where the antipodal map is now $\mathcal{V} \to -\mathcal{V} \equiv -R^2/\bar{\mathcal{V}}$ and $\mathcal{W} \to -\mathcal{W} \equiv -r^2/\bar{\mathcal{W}}$.[24] Further, $\mathcal{V}\bar{\mathcal{V}} = z^2 + y_5^2 = R^2$ and $\mathcal{W}\bar{\mathcal{W}} = x^2 + y^2 = r^2$. And after inversion,

$$z = \frac{1}{2}(\mathcal{V} + \bar{\mathcal{V}}), \qquad y_5 = \frac{1}{2i}(\mathcal{V} - \bar{\mathcal{V}}),$$
$$x = \frac{1}{2}(\mathcal{W} + \bar{\mathcal{W}}), \qquad y = \frac{1}{2i}((\mathcal{W} - \bar{\mathcal{W}}),$$
$$\varphi_1 = i \log \sqrt{\frac{\bar{\mathcal{W}}}{\mathcal{W}}} \qquad \varphi_2 = i \log \sqrt{\frac{\bar{\mathcal{V}}}{\mathcal{V}}}. \tag{77}$$

We can write

$$dz^2 + dy_5^2 + dx^2 + dy^2 = d\mathcal{V}d\bar{\mathcal{V}} + d\mathcal{W}d\bar{\mathcal{W}}. \tag{78}$$

We now have $|\mathcal{V}|^2 + |\mathcal{W}|^2 = x^2 + y^2 + z^2 + y_5^2 = r^2 + R^2$. We identified $\mathbb{C}^1 \times \mathbb{C}^1$ with $\mathbb{R}^4$, and it contains $S^3$, given by $|\mathcal{V}|^2 + |\mathcal{W}|^2 = const$. Every line through the origin, represented by $(\mathcal{V}, \mathcal{W})$, intersects the sphere $S^3$. For example, $(\lambda\mathcal{V}, \lambda\mathcal{W})$ with $\lambda = \frac{1}{\sqrt{|\mathcal{V}|^2 + |\mathcal{W}|^2}}$. Thus, the homogeneous coordinates can be restricted to $|\mathcal{V}|^2 + |\mathcal{W}|^2 = 1$. The point $(\mathcal{V}, \mathcal{W}) \in S^3 \subset \mathbb{C}^1 \times \mathbb{C}^1$ with $|\mathcal{V}|^2 + |\mathcal{W}|^2 = 1$ becomes, through complexification, a point on $S^2$, with the single complex coordinate $\mathcal{Z} = \frac{\mathcal{V}}{\mathcal{W}}$. We now have a map $H : S^3 \to S^2$ that is continuous.

One calls this a Hopf map. For each point on $S^2$, the coordinate $(\mathcal{V}, \mathcal{W})$ is non-unique because it can be replaced with $(\lambda\mathcal{V}, \lambda\mathcal{W})$, where $|\lambda|^2 = 1$ and $\lambda \in S^1$. We will now denote $\mathbb{C}_1$ as $S^2/\{\infty\}$ and $\mathbb{C}_2$ as $S^2/\{0\}$ and admit the coordinates $\mathcal{Z}$ and $\mathcal{Z}' = \frac{1}{\mathcal{Z}}$, respectively[25].

If we write

$$\mathbf{H} = \begin{pmatrix} z + iy_5 & x + iy \\ -x + iy & z - iy_5 \end{pmatrix} = \begin{pmatrix} \mathcal{V} & \mathcal{W} \\ -\bar{\mathcal{W}} & \bar{\mathcal{V}} \end{pmatrix}$$

with $det(\mathbf{H}) = |\mathcal{V}|^2 + |\mathcal{W}|^2$ and $\mathbb{R}^4 \cong \mathbf{H}$, we can also describe the mapping with normalized quaternions $\mathbb{A} = [a, \mathbf{A}] = [\cos\alpha, \sin\alpha\mathbf{n}]$ as binary rotations [51]. This becomes clearer when we consider the covering groups for covariance, such as $SO(3)$ and $SU(2)$. A rotation is effectively represented by two antipodal points on the sphere (a kind of double cover) and as a point on the hypersphere in 4D. Each rotation represented as a point on the hypersphere is matched by its antipode on that hypersphere. The quaternion represents a point in 4D. Constrained to unit magnitude, it yields a 3D space, i.e., the surface of a hypersphere. The group of unit quaternions $S^3 \subset \mathbf{H}$ is isomorphic to $SU(2)$. We identify $S^3$ and $SU(2)$ with this isomorphism. The relation with the Möbius group $G$ is then easily made (see Toth [52] or Slagter [25]), and also with the alternating group $\mathcal{A}_5$, which is isomorphic to the binary symmetry group of the icosahedron. Because the icosahedron can be circumscribed by five tetrahedra, they form an orbit of order five symmetry rotations of the icosahedron. These symmetries are subgroups of the icosahedron symmetry group. The connection with our quintic solution was made in a former study [25,26] by the observation that the vertices of the icosahedron, stereographically projected to $\mathbb{C}$, are

$$0, \infty, \gamma^j(\gamma + \gamma^4), \gamma^j(\gamma^2 + \gamma^3), \quad j = 0, ...4, \quad \gamma = e^{\frac{2\pi i}{5}}. \tag{79}$$

Briefly stated, for the icosahedral Möbius group, through a suitable orientation of the axes, we can express the linear fractional transformations as $\zeta \to \gamma^m\zeta(m = 0...4)$. The binary icosahedral group $G^*$ is associated with the Möbius group $G \subset \mathcal{M}_0(\mathbb{C})$[26].

Now, let us attempt to establish a connection with state vectors in $\mathbb{C}^2$. On a complex Hilbert space $\mathcal{H}$, which is taken as the space $\mathbb{C}^2$ consisting of pairs $|\mathcal{U}\rangle = (\mathcal{V}, \mathcal{W})$ and is equipped with a scalar product, one can define a state vector as the set of multiples $\lambda|\mathcal{U}\rangle$, with $|\lambda| = 1$, and satisfying the normalization condition $\langle\mathcal{U}|\mathcal{U}\rangle = 1$. Further, we have the matrix

$$\rho := |\mathcal{U}\rangle\langle\mathcal{U}|, \quad \rho_{ij} = \begin{pmatrix} \mathcal{V}\bar{\mathcal{V}} & \mathcal{V}\bar{\mathcal{W}} \\ \mathcal{W}\bar{\mathcal{V}} & \mathcal{W}\bar{\mathcal{W}} \end{pmatrix}, \tag{80}$$

with $\rho^\dagger = \rho$ and $\mathbf{Tr}(\rho) = 1$. We now define the vector $\vec{\mathbf{V}} = (V_1, V_2, V_3) \in \mathbb{R}^3$, with $\rho_{11} = \frac{1}{2}(1 - V_3), \rho_{22} = \frac{1}{2}(1 - V_3)$, and $\rho_{21} = \bar{\rho}_{12} = \frac{1}{2}(V_1 + iV_2)$. We can write $\rho = \frac{1}{2}(\mathbb{I} + \vec{\mathbf{V}}.\sigma)$, where $\sigma$ represents the Pauli matrices. Further, $1 - \mathbf{V}^2 \geq 0$. We define the vector $\vec{\mathbf{X}} = (x, y, z, y_5)$ and take $\lambda = e^{i\alpha}$ as a phase factor. We then write $\mathcal{W}_i \to e^{i\alpha}\mathcal{W}_i, \mathcal{V}_i \to e^{i\alpha}\mathcal{V}_i$. So, $\vec{\mathbf{X}} \to \cos(\alpha)\vec{\mathbf{X}} + \sin(\alpha)J\vec{\mathbf{X}}$, where $J\vec{\mathbf{X}} = (-X_2, X_1, -X_4, X_3) = (-y, x, -y_5, z)$ is a normalized combination of two orthogonal unit vectors in $\mathbb{R}^4$ for all $\alpha$. In fact, we obtain a Hopf fibration of $S^3$. This fibering involves the stereographic projection of $X_4 = y_5 = 0$ onto $S^3$, analogous to the stereographic projection in one dimension lower. Remember that we have identified the antipodal points, so we obtained a Klein bottle $\mathbb{K} = \{(\mathcal{V}, \mathcal{W}); |\mathcal{V}|^2 - |\mathcal{W}|^2 = \cos(\theta)\} \in S^3 \subset \mathbb{C}^2$ within $\mathbb{R}^3$ (see Equation (81) below). However, it is embedded within our $\mathbb{R}^4$. If we project from $(0, 0, 0, \pm y_0)$, the running point $\vec{\mathbf{X}}$ on $S^3$ is now related to its stereographic image $\vec{\mathbf{x}} = (x_1, x_2, x_3)$ by $X_4 = \pm\frac{x^2-1}{x^2+1}, X_i = \frac{2x_i}{x^2+1}$. For the normalization of $|\mathcal{Z}\rangle$, we first define $\varphi = \varphi_2 - \varphi_1$ and write

$$\mathcal{W} = \cos(\frac{\theta}{2})e^{i\varphi_1} = \cos(\frac{\theta}{2})e^{-i\varphi/2}e^{i(\varphi_1+\varphi_2)/2}$$
$$\mathcal{V} = \sin(\frac{\theta}{2})e^{i\varphi_2} = \sin(\frac{\theta}{2})e^{i\varphi/2}e^{i(\varphi_1+\varphi_2)/2}. \tag{81}$$

Varying $\alpha$ will only change $(\varphi_1 + \varphi_2)/2$ and not $\varphi$ and $\theta$. Further, we have $V_3 = \cos(\theta)$ and $V_1 + iV_2 = \sin(\theta)e^{i\varphi}$, $\mathcal{Z} = \frac{\mathcal{V}}{\mathcal{W}} = \tan(\theta/2)e^{i\varphi}$, with the antipodal identification $\varphi \to \varphi + \pi$. One can compare this visualization to the orientable counterpart situation of the torus (for details, see Toth [52] (Section 1.4) or Urbantke [50]). So, if we consider a Hawking particle falling in, it travels for a while in $\mathbb{R}^4$, transitioning from $y_5 \to -y_5$, and arrives at the antipodal point in $\mathbb{R}^3$.

*5.3. Treatment of the Gravitational Backreaction*

The most ideal procedure is to solve the system of PDEs with the $(\theta, \varphi)$ coordinates (or $\varphi$ alone in our case). This approach was carried out in the case of the Vaidya spacetime using the high-frequency method [45,46]. However, one can gain some insights from an approximation procedure. This involves temporarily switching off the gravitational backreaction by considering "soft" particles. In short, a firewall caused by ingoing particles can be replaced with outgoing particles. So, the entire distribution of all ingoing particles is related to the entire distribution of the outgoing ones. In this approximation, one obtains a black hole evolution operator. The antipodicity is mandatory (in our case, the Klein surface); otherwise, the relation between ingoing and outgoing particles would not be unitary. Here, quantum mechanics comes into play: the relation can be described by the relation between momenta and position; one is the Fourier transform of the other. The transform is unitary if one incorporates regions I and II of the Penrose diagram, i.e., integration from $-\infty$ to $+\infty$. So, the process is time-reversible. The firewall transformations come together with the so-called *Shapiro* shift effect. The gravitation interaction is described by this shift effect, a time delay of signals grazing past a heavy object. One uses in the calculation of the expansion in spherical harmonics. One states that the Hawking particles, which originated near the horizon with very high energy and momentum, can be regarded as decaying firewalls. What about the entanglement? As already stated, one also has to average over the contribution of the quantum states of region II. However, the inside observer will not detect these states. We cannot include these states in the final stage of the evaporating black hole. So, pure quantum states turn into mixed states. One could conclude that the information from the states $|E, n\rangle_{II}$ will eventually escape from the black hole in the distant region III. This leads to the well-known paradoxes. We already explained the solution, i.e., through the firewall transformation and the antipodal map. The information retrieval process is summarized as follows: the outgoing particle carries the information of the ingoing particle back to the outside world (via the shift to the new coordinates of the outgoing particles). So, a firewall will not build up[27]. To see that the antipodal map is mandatory, we first note that there was no one-to-one mapping of the coordinates in regions I and II. The map is one-to-two. Are there two black holes? An Einstein–Rosen bridge does not solve the problem (or another universe). The antipodal map makes the mapping one-to-one, yet they remain distinct because they are space-like separated. The states $|E, n\rangle_{II}$ and $|E, n\rangle_{I}$ now describe visible particles outside the horizon. This means that the state no longer requires summation over unseen states to form the density matrix. We still have a pure state. If one sums over the other states far away from the antipodal area and unobservable to the local observer, then one would conclude that these particles have the Hawking temperature. However, the entire state is not thermal. The Hawking particles on one hemisphere are entangled with the antipode. For the outside observer, the Hartle–Hawking wave function is no longer a thermally mixed state, but a single pure state. In general, the antipodes are completely entangled. So, the outside observer is far from the black hole, and the black hole is not in a stationary Hartle–Hawking quantum state. Or to state it differently, there is no complete heat bath. This does not violate Einstein's equivalence principle because the points at the opposite sides of the horizon are always space-like separated. It seems that the particles entering and emerging from the black hole can be seen as quantum clones in regions III and IV. This is a point of concern. In our model, we shall see that this concern disappears. In Section 4, we mentioned the time delay in the Hawking particle traveling on the Klein surface in our model. If this time is

detectable, it could provide evidence of the antipodal map, as well as evidence of an extra dimension. A second point of concern is the infinitely fast transport of information from a point at the horizon to its antipode. One can argue that for the outside observer, this lasted an infinitely long time in coordinates (U,V). In cosmological time, in our model, this transport will take a finite time, possibly measurable. Note that for the co-moving observer, the Hawking particles continue their way falling inwards, and they do not observe the re-emission. A third point of concern is the time reversal in region II. Could this time reversal be related to thermodynamical entropy? This is not necessary, as proposed by Lloyd [53]. Maybe the degree of entanglement determines the arrow of time. Quantum uncertainty and the way it spreads as particles become increasingly entangled could replace the standard notion of the arrow of time. The direction of time can thus be related to the increase in quantum mechanical correlation. The particles running backward in region II decrease in quantum mechanical correlation[28]. To quote Lloyd: "The universe as a whole is in a pure state, but individual pieces of it are in a mixture because they are entangled with the rest of the universe. When you read a message on a paper, your brain becomes correlated with it through the photons that reach your eyes. At that moment, you will be capable of remembering what the message says. So the present can be defined by the process of becoming correlated with our surroundings". Finally, the approximation of the gravitational dragging needed to describe the transfer of the information and bypass the problem of the non-constant background metric is only valid for a small time interval. Just like in the Regge–Wheeler approximation, there is the problem of matching. In our model, we still have the dynamical equations on the effective 4D spacetime. Our decoupling of the dynamical part and the spherical harmonics is not performed manually but follows from the PDEs.

*5.4. Treatment of the Quantum Fields*

An outside observer will register the Hawking radiation as thermal, i.e., in a mixed state, whereas a local observer will be in doubt about the vacuum state. Further, the evolution of the wave function of the infalling particle must be unitary, i.e., it satisfies the Schrödinger equation $|\chi(t_1)\rangle = U(t_1, t_2)|\chi(t_2)\rangle$ and is bijective. It is believed that during black hole evaporation, information on the quantum state is preserved. Information loss is inconsistent with unitarity. So, the problem is how to handle the controversy between the pure quantum state of the infalling particles and the mixed state property of Hawking radiation. From the antipodal mapping in $\mathbb{R}^4$, we also have $e^{in\varphi} \to e^{in(\varphi+\pi)}$. This can be "visualized" by examining Figure 4, where we consider points on the circle on $S^3$ for which

$$\sin(\theta/2)e^{in\varphi}\mathcal{V}(\alpha) - \cos(\theta/2)\mathcal{W}(\alpha) = 0. \tag{82}$$

From the real part, we obtain the plane after the stereographic projection,

$$x_3 = \tan(\theta/2)(\cos(\varphi)x_1 - \sin(\varphi)x_2). \tag{83}$$

If we apply $e^{in\varphi} \to e^{in(\varphi+\pi)}$, the plane is rotated over $\pi$. The imaginary part delivers the two spheres $\mathbf{x}^2 \mp \tan(\theta/2)(\sin(\varphi)x_1 + \cos(\varphi)x_2) = 1$. In the following, we propose that the infalling Hawking particle travels for a while on the Klein surface in $\mathbb{R}^4$. Now, consider the state

$$|\chi\rangle = |\hat{\chi}\rangle e^{in\varphi} \tag{84}$$

and the density matrix

$$\rho_A = \sum_i p_i |\chi_i\rangle_{AA}\langle\chi_i| \tag{85}$$

as a mixture of pure states and weights $p_i$, where $\sum p_i = 1$. Further, $\mathbf{Tr}\rho_A = 1$ and $\rho_A^2 = \rho_A$. The density matrix does not contain the phase. So, the change in the azimuthal angle

$\varphi$ has no influence on the pure states. This can be compared to the "pureness" of $\rho$ in Equation (80) of normalized state vectors on $S^2 \subset \mathbb{R}^3$. In addition to $\vec{\mathbf{V}}$,

$$\vec{\mathbf{V}} = \left( \mathcal{V}\bar{\mathcal{W}} + \mathcal{W}\bar{\mathcal{V}}, i(\mathcal{V}\bar{\mathcal{W}} - \mathcal{W}\bar{\mathcal{V}}), |\mathcal{V}|^2 - |\mathcal{W}|^2 \right), \tag{86}$$

independent of a phase change in $|\mathcal{U}\rangle = (\mathcal{V}, \mathcal{W})$, and we define

$$\vec{\mathbf{Z}} = \vec{\mathbf{P}} + i\vec{\mathbf{Q}} = \left( \mathcal{V}^2 - \mathcal{W}^2, i(\mathcal{V}^2 + \mathcal{W}^2), -2\mathcal{V}\mathcal{W} \right). \tag{87}$$

$(\vec{\mathbf{V}}, \vec{\mathbf{P}}, \vec{\mathbf{Q}})$ forms a positively oriented orthonormal triad. Now, $\vec{\mathbf{P}}$ is a unit tangent vector to $S^2$ at $\vec{\mathbf{V}}$. If $|\mathcal{U}\rangle \to e^{i\alpha}|\mathcal{U}\rangle$, then $\vec{\mathbf{V}}$ remains independent of this phase change, whereas $\vec{\mathbf{Z}} \to e^{2i\alpha}\vec{\mathbf{Z}}$. One can verify that $\vec{\mathbf{P}}$ registers a phase change $mod(\pi)$. When $|\mathcal{U}\rangle \to -|\mathcal{U}\rangle$, $\vec{\mathbf{U}}$ still yields the same $\vec{\mathbf{P}}$, whereas $(\vec{\mathbf{R}}, \vec{\mathbf{P}})$ together determine $|\mathcal{U}\rangle$ up to sign. Let us now interpret $|\langle \mathcal{U}|\mathcal{U}'\rangle|^2$ (independent of the phase) as a "probability". In the language of the geometric picture, here, $\sqrt{1 - |\langle \mathcal{U}|\mathcal{U}'\rangle|} = \frac{1}{2}|\vec{\mathbf{V}} - \vec{\mathbf{V}}'|$ represents the diameter of $S^2$. This "visualization" is in fact a Hopf fibration of $S^3$ [52]. In our case, we are dealing with a time-dependent situation, and the state vectors follow the Schrödinger equation

$$\frac{d}{dt}|\mathcal{U}\rangle(t) = -\frac{1}{\hbar}H|\mathcal{U}\rangle(t), \tag{88}$$

where $H$ is a Hermitian matrix, with $H = H_0\mathbb{I} + \vec{\mathbf{H}}.\sigma$. One finally obtains

$$\frac{d\vec{\mathbf{V}}}{dt} = \dot{\vec{\mathbf{V}}} = \frac{2}{\hbar}\vec{\mathbf{H}} \times \vec{\mathbf{V}}, \tag{89}$$

which is the time evolution in $S^2$. From Equation (86) we have $\dot{\vec{\mathbf{V}}}.\vec{\mathbf{V}} = 0$, so a pure state remains pure and a mixed state remains mixed. In the former antipodal description on the hyper three-surface, it was concluded that a local observer sees a vacuum, the Hartle–Hawking state

$$|0\rangle_{HH} = C \sum_{E,n} |E, n\rangle_I |E, n\rangle_{II} e^{-\frac{1}{2}\beta E} \tag{90}$$

He can use the creation and annihilation operators. All these states remain pure due to the antipodal identification. However, the loose entanglement of the HH state exists. So he observes a perfect thermal mixture with Hawking's temperature. The outside observer concludes that the vacuum contains states $|E, n\rangle$ in regions I and II of the Penrose diagram. His Hawking particles are not precisely thermal since there is strong entanglement between the particles emitted at the opposite hemispheres of the horizon. Further, region II is not "really there". It represents the inside of the black hole. The only problem is that the information seems to travel infinitely fast to the antipodal point.

　　In our warped spacetime, we noticed that the cut-and-pasted identification can be replaced with the Klein surface: there is no need for instantaneous transport of information. Further, while traveling in the bulk for a while in local time (or in the tortoise coordinate system) close to the brane, the gravitons become "hard", i.e., very massive, due to the shape of the gravitational probability function. This causes backreaction on the brane (Section 2). Moreover, when returning to the brane, we do not encounter problems concerning the firewall caused by the hardness of the particles. In Figure 5, we visualized the 5D topology.

　　The two antipodal regions $(I + III)$ and $(II + IV)$ are marked on the brane. Close to the center, quantum effects emerge. We have $UV = (r - r_H)^{\frac{\kappa r_H}{2(r_H + b_2)^3}}$ (see also Figure 1). Arrows in red represent the ingoing particles, whereas those in green represent the outgoing particles. The ingoing Hawking particles reappear at the antipode. However, they stay awhile on the Klein surface. Again, the particles remain entangled. The embedding of the Klein bottle is without self-intersection in $\mathbb{R}_+^2 \times \mathbb{R} \times S^1$.

Finally, we make a remark on the role of the dilaton. The gravitational force is small in the effective 4D spacetime. This is due to the warp factor, which is the amount by which one rescales the energy. On the weak brane, the graviton has a small probability function, while away from the brane, the gravitational force "spreads" into the extra dimension and becomes heavier [11]. We have already noticed that the dilaton field describes the complementarity. The local observer can rescale his $\omega$. According to our definition of spacetime in Equations (4), (9), and (12), we have two contributions. They both behave as scalar fields on small scales. The 5D part could be the radion field (or "gravi-scalar"), which characterizes the RS-1 model. In Section 2, we found that the $\mathcal{E}_{\mu\nu}$ tensor carries information about the gravitational force outside the brane. We also found that it affects the evolution of the brane fields, which delivered the new exact solution. Therefore, we do not need a perturbation method to find the $y_5$-dependent part. Our conformal model provides a constant value for $\omega(y_5)$. So, $\mathcal{E}_{\mu\nu}$ is the only imprint of the bulk[29]. Further, we replace the Shapiro effect with the interaction of the dilaton field and the " unphysical" metric $\tilde{g}_{\mu\nu}$, along with the scalar field. This is plausible due to the curvature of the Klein surface, which is conformally flat. In the RS2 model, we have the $\mathbb{Z}_2$ symmetry. So, the mapping $y_5 \to -y_5$ identifies the Klein surfaces. One could say that the transition to the antipode map in the extra dimension becomes less "mysterious". Initially, the Hawking particles traverse a proper finite time. The apparent discontinuity during the "jump" disappears in the Klein model within the RS2 warped spacetime model.

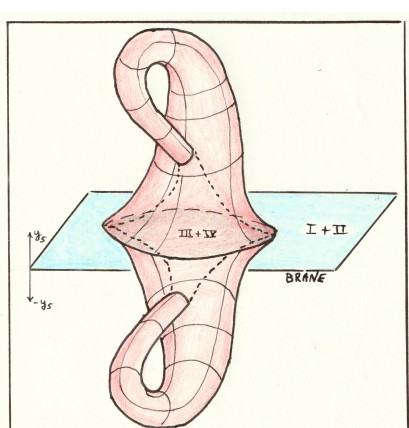 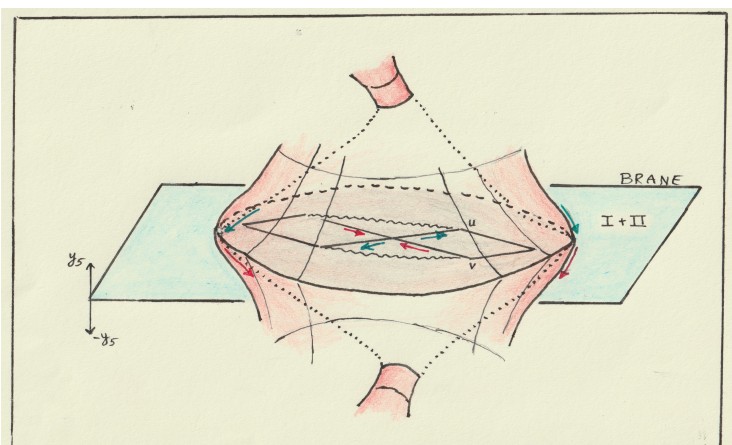

**Figure 5.** In the RS2 model (1 brane), there is a mirror symmetry $\mathbb{Z}_2$, i.e., $y_5 \to -y_5$. So, we have a contribution from the antipode when considering Hawking radiation (see text). The antipodal map on the brane is suppressed for clarity. Right: the ingoing and outgoing Hawking particles.

## 6. Conclusions

We can formulate the main results of our model as propositions:

**Proposition 1.** *The vacuum solution found in the conformal invariant 5D warped spacetime, where the metric is determined by a quintic polynomial, can be described by a Laplace-type equation on a Klein surface. This quintic polynomial can have one real zero with a multiplicity of 5 under some constraints of the parameters of the model under consideration. On the Klein bottle, the first non-zero eigenvalue of the Laplacian also has a multiplicity of 5. The minimal area is determined through an elliptic integral of the second kind.*

**Proposition 2.** *The antipodal mapping of regions I and II in the Penrose diagram is accompanied by the continuous transition in the extra bulk dimension using a non-orientable Klein surface that is embedded in $\mathbb{R}^4 \sim \mathbb{C}^1 \times \mathbb{C}^1$.*

**Proposition 3.** *In the conformal dilaton-scalar field model on a warped spacetime, the field equations for the dilaton $\omega$, which is part of the gravitational part of the spacetime, and the scalar*

*field obey comparable Klein–Gordon-type PDEs. The interaction of these quantum fields in the "un-physical" conformally flat spacetime describes the time evolution of the horizon.*

**Proposition 4.** *The information problem was solved by the antipodal boundary condition, i.e., by removing the inside of the black hole. The result was, that the black hole is not thermal because there is no "hidden sector" to produce the thermodynamically mixed state of the Hawking radiation. All states remain pure for the outside observer.*

*In our model, the antipodal map on the brane is accompanied by the $\mathbb{Z}_2$ symmetry in the bulk in the RS2 model. This means that there is no infinitely fast transport to the other side of the hemisphere. The Hawking particles remain for a while on the Klein surface, as seen by the outside observer.*

**Proposition 5.** *The geometric quantization of our Klein surface can be constructed using a two-level Hilbert space on the complex hypersurface $\mathbb{C}^1 \times \mathbb{C}^1$.*

## 7. Discussion

One can conjecture that new Planck scale physics should be necessary to address the black hole paradoxes, such as the information paradox. Considering only the low energy seems inadequate for describing Hawking radiation together with gravitational interaction. The Hilbert space will be affected, and one needs new geometrical boundary conditions to incorporate the quantum effects. One of the issues that needs clarification is the physical interpretation of the maximally extended Schwarzschild spacetime manifold in Kruskal–Szekeres coordinates. The geometry of the spacetime can be divided into four regions in the Penrose diagram. An alternative method is the "cut-and-paste" method, where the "obscure" regions II and IV are the antipodes of our real regions I and III. One secures the unitarity of the S-matrix. The handling of high energies and the gravitational backreaction is performed in this model through the Shapiro dragging by considering the equations in the transverse spherical coordinates. One usually expands the spherical harmonics in $\mathbf{Y}_{l,m}(\theta, \varphi)$ and the dynamics factorizes in $(l, m)$. Soft particles transform into hard particles for a short time, removing the firewall. Region II then refers to the same black hole as region I, but for the solid angle, we apply the map $(\theta, \varphi) \to (\pi - \theta, \varphi + \pi)$ together with the dynamical map of the Kruskal–Szekeres coordinates $(U, V) \to (-U, -V)$.

We presented a solution of a conformal invariant dilaton–Higgs Kerr-like black hole on a 5D warped axially symmetric spacetime and clarified some shortcomings of the antipodal mapping. We replaced the spherical harmonics with cylindrical ones $\mathbf{Y}_m(\varphi)$ and used the winding number of the scalar-Higgs field as a second quantum number. The dynamical part was obtained using partial differential equations in $(r, t)$ and could be solved exactly in the vacuum case. It was conjectured that the angle-dependent part of the model could be derived in the non-vacuum case to confirm the numerical solution. By applying conformal invariance, we could treat the small-scale dynamics through the dilaton field, which is coupled to the scalar field. It also described the complementarity between the ingoing and outside observers. Using the Randall–Sundrum warped brane-world model with one extra dimension $y_5$, one antipodal map became $(U, V, z, \varphi, y_5) \to (-U, -V, -z, \varphi + \pi, -y_5)$, with an extra $\mathbb{Z}_2$ symmetry in $y_5$. This means that the stereographic projection of $S^3$ was replaced with $S^4 \sim \mathbb{C}^1 \times \mathbb{C}^1$, which is a Klein bottle embedded in $\mathbb{R}^4$. The Hawking particles remained pure because the in-particle stayed on the Klein surface for a short time and remained entangled. No infinitely fast transport through the antipodal map was necessary.

**Funding:** This research received no external funding.

**Data Availability Statement:** There is no data used.

**Conflicts of Interest:** The author declares no conflict of interest.

## Notes

1.      It is a global symmetry when one considers the spacetime as fixed.
2.      One can also use the Eddington–Finkelstein coordinates $(U, r, z, \varphi, y_5)$.
3.      For the time being, we omit the cosmological constant term $\sim \Lambda \omega^4$.
4.      The signs of $a_i$ and $C_i$ are not important yet.
5.      Here, we will not treat the conformal anomalies [30].
6.      In this case, the $dz^2$ term is maintained.
7.      The sign of $a_2$ is just a matter of convention.
8.      This technical aspect is currently under investigation by the author.
9.      See Slagter [26]. The replacement $t \to i\tau$ has no influence on the solution.
10.      When we apply the antipodal mapping, 't Hooft suggests cutting out an $S^3$ sphere, with topology $\mathbb{R} \times S^3/\mathbb{Z}^2$. This is not necessary in our 5D model. The effective 4D spacetime is already present.
11.      If one allows a Higgs field, then for $d = 2$, one calls the configuration a vortex, and for $d = 3$, a monopole.
12.      In Euler angles $(\theta, \xi, \varphi)$.
13.      Note that we have the double cover $\mathbb{C}^1 \times \mathbb{C}^1$ for our original spacetime. Then, $\varphi$ indeed runs from $0...2\pi$.
14.      Remember that $\omega$ contains the gravitational constant $\kappa$ due to the redefinition.
15.      The scalar-gauge system possesses a quantized magnetic flux $\sim \frac{n}{e}$, which equals the first Chern number of $A$ on $\mathbb{R}^2_+$. This has an important consequence in the expansion of the scalar field in cylindrical harmonics (see Section 5).
16.      Another interesting method could be derived from the equation of the directrix of the Klein bottle. From the integral curve of the PDEs, $\frac{\partial r}{\partial t_e}$, one could find, in principle, the elapsed time [37].
17.      There is a difference between Maldacena's method and our model: our method does not need a matter field in the bulk.
18.      There are no fixed points.
19.      Also called the "cut-and-paste" procedure or firewall transformation.
20.      Further, our slice is the Klein surface!
21.      Remember that we used twice the dilaton separation.
22.      Remember that our solution for $N$ in Section 4.2 could be written as [25] a meromorphic polynomial $\frac{P(\mathbf{z})}{Q(\mathbf{z})}$, with $Deg(P) = 5$.
23.      With $x = r \sin \varphi, y = r \cos \varphi$.
24.      Because $e^{i(\varphi + \pi)} = -e^{i\varphi}$.
25.      Let $G$ be the group of self-homeomorphisms of the product space $S^2 \times S^2$, generated by interchanging the two coordinates of any point and by the antipodal map on either factor. $G$ is then isomorphic to the dihedral group. It contains three subgroups, for example, $K = \{I, (x, y) \to (-x, y), (x, y) \to (x, -y),$ and $(x, y) \to (-x, -y)\}$. It acts freely on $S^2 \times S^2$. Thus, $(S^2 \times S^2)/K = \mathbb{R}P^2 \times \mathbb{R}P^2$. The most interesting feature is the fact that the twofold symmetric product of $\mathbb{R}P^2$, $SP^2(\mathbb{R}P^2) = \mathbb{R}P^4$.
26.      Remember, $SO(3) \cong \mathcal{M}_0(\mathbb{C}) = SU(2)/\{\pm I\}$.
27.      For a more extensive description, see t' Hooft [10].
28.      Another issue is the proposition that the entropy is proportional to the horizon area and not the volume. There is no inside in the antipodal map. In the context of the Klein surface, does the area extend to the Klein surface? In the RSII model, this is not an issue because the two mirror surfaces cancel each other.
29.      From the black hole, where we have an FLRW spacetime, the warp factor is $y_5$-dependent. One then Fourier expands the perturbative 5D graviton amplitude as $f(\mathbf{x}, y_5) = \sum_m e^{imy_5} f_m(\mathbf{x})$ The KG equation for f is replaced with our dilaton field.

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
