# Peer review of "Quantum Black Holes in Conformal Dilaton–Higgs Gravity on Warped Spacetimes"

_universe, doi:10.3390/universe9090383_

Round 1

Reviewer 1 Report

This work builds on a previous proposal by the same author (Refs [18,19]) for a closed-form metric of a 5D spacetime with a "warp-factor". The present work analyzes a black hole form of the previously proposed solution and calculates normal modes of a scalar field in this background, including a Z2 orbifold that identifies region II (the parallel universe part) of a black hole with region I (our universe).

This could be an impressive work were it not for several deficiencies in both presentation and contents. The main problem for me is that I did not understand what the proposed metric is exactly. As I understand it, the proposal is that the metric given by equation (5) solves the 5d (bulk) Einstein's equations in vacuum (with a cosmological constant) provided N(t,r) and omega(t,r) are chosen as in equation (17). If that's indeed correct, I think it is a very elegant result! However, the paper also claims that omega(t,r) needs to satisfy a "constraint equation" given by (18). I expect a constraint to be an algebraic relation among the constants that enter equation (17). But equation (18) is a partial differential equation, and it is not clear that it reduces to a constant.

Moreover, equation (18) includes the symbol bar-omega that was not defined before. It entered equation (11) and it is mentioned that (line 154) "... omega and bar-omega differ only by the different exponent 3/2 and 1 respectively", but it would be much easier for the reader if bar-omega was written explicitly. Also, (18) contains both omega and bar-omega, and I'm not sure if it's intentional or not. I checked a previous work by the same author (arXiv:2012.00409) and found that bar-omega was defined there as omega but without the exponent n/2-1, but when I plugged that into equation (18) I didn't see how it reduces to an algebraic relation among a1, a2, a3, C1, C2, C3, Lambda, y0, kappa. (I used Mathematica.) If this issue can be settled, and if indeed equations (5),(17)-(18) are an exact solution, then I think the paper should be published, after additional minor problems mentioned below are addressed.

Let me mention below additional concerns:

1. Equation (1) appears to have the wrong sign for the Einstein-Hilbert term relatively to the kinetic term of the scalar fields. I think this has to do with the rescaling mentioned on line 96 which says to replace omega2  with -6*omega2/kappa2. But the opposite sign in the coefficient of the curvature is a problem: for example, gravitational waves would carry negative energy!

2. The discussion in section 2 seems to jump from the metric g to tilde-g and from 5d to 4d, and it was hard for me to follow which metric is used for which equation. For example, I assume (5) is for g, I'm not sure why the rescaling in (11) is necessary, and on Line 152 it says n=4,5 and I'm not sure why we can't just take n=5, as in the discussion before. I beg the author to make it easier for the reader and stick with one dimension and one metric for as long as possible.

3. why don't the bulk fields omega and Phi appear in the bulk Einstein's equation (6) through their energy momentum tensor?

4. Equation (8) doesn't seem correct: the 5d metric component g55 is, according to (5), omega24/3=y04/3 while the component n5 of the normal vector is n5=(g55n5 )2=y08/3y0-1=y05/3.

5. I didn't quite understand the proposal how to resolve the ``Firewall''. As I understand it, the Firewall problem is for an old one-sided Black Hole (e.g., forming from gravitational collapse) and has to do with the problem that the Hawking radiation cannot be maximally entangled with both the old Hawking radiation and the interior of the Black Hole. But more modern approaches resolve the paradox anyway. (See for example Penington's work arXiv:1905.08255.)

6. Line 45: I'd suggest to mention specific references next to "Some physicists suggested that region II ...". Also "an other" should be "another".

As far as I can tell, the English and grammar is generally good, but there are some typos. For example:

Line 63: "in order [to] describe"

Line 88: "dynamically" should be "dynamical"

Line 112: "recovers the for ..."

Line 115: "go through it" should be "goes through it"

Line 175: "When on[e] omits"

Line 396: "unitairity"

Author Response

I unloaded my reply.

Where can I upload the revised manauscript?

Reviewer 2 Report

see att

see peer-review-30707942.v1.pdf

Author Response

I uploaded my reply

Where can I upload the revised version?

Reviewer 3 Report

I think the paper is really well presented and very clear in all their main concepts/ideas. It should be published in its current state.

Author Response

No reply, because the reviewer agrees

Round 2

Reviewer 1 Report

I'd like to thank the author for including additional explanations. They are helpful!

I am still unsure about the role of the "extra equation" [equation (20) on page 6 of the revised version]. I used the solution from equations (19) and I don't see how equation (20) is satisfied by that solution.

I also still don't understand why the sign in front of the curvature term in equation (1) is not the wrong one. Usually, the coefficient of the curvature term is taken positive, but it is clearly negative in (1). Conformal gravity would add higher order corrections (like the square of the Weyl tensor) to the action, but those are irrelevant at low-energy, and if there is a negative sign in front of the Einstein-Hilbert term, long-wavelength gravitational waves will carry negative energy. [See equation (4.76) of gr-qc/9712019, for example, for the standard sign.]

Another issue that I'd like to ask about and wasn't in my original report is the Penrose diagram (Fig 1). Won't the other roots ri of the quintic polynomial in equation (32) correspond to "inner horizons" that will modify the form of the Penrose diagram, similarly to what happens for a Reissner-Nordström black hole where the metric factor is quadratic? For example,  it seems to me that there is a limit of the proposed metric which reduces to a static black hole. (Take a3 compared to the rest of the parameters.)

Below is a short list of Mathematica commands that I used to check equation (20). Perhaps there is an error in the file, or I misunderstood the formulas, in which case I'd be delighted to amend my recommendation.

(* Mathematica commands to chek Eqn (20) *)

fN = n[t, r]; (* the function N *)

fOmegaBar = oBar[t, r]; (* the function omega-bar *)

Eqn17 = -D[fOmega, {t, 2}] - fN^4*D[fOmega, {r, 2}] + (5/3/fOmega)*(fN^4*D[fOmega, r]^2 + D[fOmega, t]^2);

Eqn18 = -D[fN, {t, 2}] + 3*D[fN, t]^2/fN -fN^4*(D[fN, {r, 2}] + 3*D[fN, r]/r + D[fN, r]^2/fN) - (2/fOmega)*(fN^5*(D[fOmega, {r, 2}] - 5*D[fOmega, r]^2/3/fOmega + D[fOmega, r]/r) + fN^4*D[fOmega, r]*D[fN, r] + D[fOmega, t]*D[fN, t]);

(* The following are the proposed solution from Equations (19) *)

a[4] = a[2]*a[3];

o[t_, r_] := (a[1]/((r + a[2])*t + a[3]*r + a[4]))^(3/2);

oBar[t_, r_] := (a[1]/((r + a[2])*t + a[3]*r + a[4]));

n[t_, r_] :=Sqrt[(1/5/r^2)*(10*a[2]^3*r^2 + 20*a[2]^2*r^3 + 15*a[2]*r^4 + 4*r^5 + c[1])/(c[2]*(a[3] + t)^4 + c[3])];

nValue = 4;

Eqn20 = -D[fOmegaBar, {r, 2}] - L*fOmegaBar^((nValue + 2)/(nValue - 2))/fN^2 - D[fOmegaBar, r]*D[fN, r]/fN - D[fOmegaBar, r]/2/r + 4*D[fOmegaBar, t]^2/fOmegaBar/fN^4/(nValue - 2) -D[fOmegaBar, t]*D[fN, t]/fN^5;

(* Checking equation 20 *)

W = Simplify[Eqn20,  Assumptions -> {r > 0, a[2] > 0, a[1] > 0, a[3] > 0, L > 0, t > 0}]; (* Doesn't give something that is obviously ZERO *)

Simplify[W /. {t -> 0, a[2] -> 0}]; (* still doesn't give zero *)

Author Response

Attached please find my responses.

Reviewer 2 Report

The paper can be published in the present form up to a short remark about the "the island recipe".

In fact, the author ask what is "the island recipe".

The island  prescription help  to solve the information paradox for certain  regions.

``The island recipe'' was introduced to explain the stoppage of the entanglement entropy growth, see Penington (1905.08255 [hep-th]),

Almheiri (1905.08762 [hep-th]), 

Almheiri (1908.10996 [hep-th])

I suggest to the author to look on above mentioned papers and make some remarks about this proposal in the text

Author Response

I send you the review #2

Round 3

Reviewer 1 Report

I thank the author for explaining that there are no additional real roots that could modify the Penrose diagram.

But I'm afraid I must insist about the first two points in my last report.

1. I understand that the author agrees that the solution presented in equation (19) does not solve equation (20). But if equation (19) fails to satisfy any equation of motion, then it is not a solution. So, I don't understand why it is presented. And in any case, I think it should be clearly stated that (19) doesn't satisfy (20).

2. In appendix B of arXiv:1405.1548, the sign of the standard Einstein-Hilbert action is positive. To that EH action R/16*pi*G is then added the conformally coupled scalar with its coupling -phi^2*R, but I believe the intention is to study it in the region 16*pi*G*phi^2<1 where the overall factor in front of the curvature is still positive.

In equation (1) of the present paper, however, there is no R/16*pi*G, so the overall factor in front of the curvature is always negative, and therefore I suspect the system will be unstable (e.g., gravitational waves carry negative energy and their emission will increase the energy of the emitter).

Author Response

Reply # 3 on reviewer #1
I am very grateful again to the reviewer for the valuable comments.
I have made some remarks and some additions.
The corrections/additions are done in RED.
1. I added extra information about the special vacuum solution.
2. I Agree. We redefined ω2 →-6ω2/κ2 (see below Eq.(3))
Then we put outside the action the κ term. So it enters then in the potential term. See the added
explanation just after Eq. (1).
Concerning the gravitational wave emitted:
We wrote the spacetime as the product of ω2 times the “un-physical” g, which is conformally flat.
I added a rather extensive explanation of the use of the ω.
All the emission with be due to ω2 , which is positive. This occurs only close to the horizon.
Best regards,
R. Slagter
